# A Cone Fishway Facilitates Lateral Migrations of Tropical River-Floodplain Fish Communities

**Lee J. Baumgartner [1,2,*]**, **Craig Boys [3]**, **Tim Marsden [4]**, **Jarrod McPherson [2]**, **Nathan Ning [2]**, **Oudom Phonekhampheng [5]**, **Wayne Robinson [2]**, **Douangkham Singhanouvong [6]**, **Ivor G. Stuart [7]** and **Garry Thorncraft [5]**

1 Narrandera Fisheries Centre, New South Wales Department of Primary Industries, PO Box 182, Narrandera, NSW 2700, Australia
2 Institute of Land, Water and Society, Charles Sturt University, PO Box 789, Albury, NSW 2640, Australia; jmcpherson@csu.edu.au (J.M.); nning@csu.edu.au (N.N.); wrobinson@csu.edu.au (W.R.)
3 Port Stephens Fisheries Institute, New South Wales Department of Primary Industries, Locked Bag 1, Nelson Bay, NSW 2315, Australia; craig.boys@dpi.nsw.gov.au
4 Australasian Fish Passage Services, 6 Spinifex Street, Fern Bay, NSW 2295, Australia; Tim.Marsden@ausfishpassage.com
5 Department of Livestock and Fisheries, National University of Laos, PO Box 7322, Vientiane, Lao PDR; oudomg@yahoo.com (O.P.); garrythorncraft@yahoo.com.au (G.T.)
6 Living Aquatic Resources Research Centre, PO Box 9108, Vientiane, Lao PDR; douangkhams@gmail.com
7 Kingfisher Research, 177 Progress Road, Eltham, VIC 3095, Australia; ivor.stuart@gmail.com
* Correspondence: lbaumgartner@csu.edu.au

**Abstract:** Fisheries in many tropical river-floodplain systems are under threat from physical obstructions caused by ongoing river infrastructure development. There is a growing need for innovative, cost-effective technologies to mitigate the impacts of these obstructions. This study examined the effectiveness of a new cone fishway for facilitating lateral migrations of river-floodplain fish communities in the Lower Mekong Basin in Lao PDR. We assessed the species richness, size range, abundance and biomass of fish able to pass through a cone fishway, using paired entrance and exit sampling during both dawn/day and dusk/night. Overall, a diverse range of taxa (76 species) and size classes (25–370 mm) ascended the cone fishway. The total size range of fishes observed at the fishway entrance was similar to that at the exit, although the fish at the entrance were significantly smaller (in length) than those at the exit, during both diel periods. Additionally, there were significantly higher abundances of fish at the entrance than at the exit, but there was no difference in total biomass, again for both periods. These results suggest that, with further development, the cone fishway design has considerable potential for facilitating the lateral migrations of diverse tropical river-floodplain fish communities at low/medium head infrastructure.

**Keywords:** fish passage; Lower Mekong Basin; river infrastructure; sustainable irrigation; wetland

## 1. Introduction

Tropical river-floodplain systems support the most diverse and productive freshwater fisheries in the world, yet paradoxically they are being increasingly exploited to meet growing food and energy requirements [1,2]. River management infrastructure, such as dams and floodplain regulators, are proliferating in many tropical river-floodplain systems to meet growing global demand for irrigated agricultural resources and power generation [3]. Whilst river infrastructures have been crucial to advancing agricultural and energy production, they have created physical barriers to spatially separate fish spawning, nursery and feeding habitats, and subsequently prevented fish from completing their

life cycles [1,4]. Consequently, there have been major declines in the productivity and diversity of the fisheries in many tropical river-floodplain systems, and these declines are likely to intensify over the coming decades, with localised extinctions expected [2,5].

Fishways have been installed in many tropical river-floodplain systems in an effort to improve fish passage where river infrastructure blocks migratory routes [6,7]. Most early fishways in tropical river-floodplain systems were based on designs used for salmonids in temperate systems (e.g., pool-and-weir fishways) [8]. Unfortunately, these fishways typically failed, owing to inherent differences in the movement ecology of temperate salmonid fishes and tropical non-salmonid species [9,10]. In the last 20 years, there has been a shift towards the use of low-turbulence vertical slot fishways, which operate over a broad hydrological range [6,7,11]. Such fishways have had reasonable success in improving fish passage in tropical rivers (e.g., the Mekong River in South East Asia [6,11]) and coastal subtropical rivers (e.g., the Burnett River in Australia [8]). However, with increasing water scarcity, vertical slot fishways, like many pool-and-weir type fishways (e.g., submerged orifice designs), may unintentionally drain headwater pools if their slots/orifices are below the minimum headwater level [12]. In countries with low construction standards, fishways with deep pools and restricted access may also present a potential danger for human drownings, especially in remote areas where subsistence fishers and children regularly interact with infrastructure [12]. Consequently, there is a pressing need for continued development of existing and innovative fishway designs which work effectively and also fulfil community and safety-in-design standards.

Recently, a cone fishway design was developed in an attempt to provide an alternative to more traditional pool-type fishway designs for tropical river-floodplain system fishes [13–15]. The cone fishway is based on rock ramp design principles (i.e., where there are multiple ascent pathways rather than a single slot), and has a fixed crest level formed by precast concrete cone-shaped baffles, making it best suited to sites where there is a narrow headwater range (e.g., <0.4 m) [13,15]. In addition to offering an alternative to other pool-type fishways (e.g., vertical slot) with respect to fish passage performance, the cone fishway: (1) is simple in design and low maintenance [15]; (2) can potentially be built from pre-fabricated baffles to reduce the overall construction costs [15]; (3) offers the potential to have lower average turbulence (energy dissipation) than the vertical slot fishway [14]; (4) prevents a headwater pool from being completely drained due to its fixed crest level [13]; and (5) offers potentially safe human access and egress to better meet safety standards [12].

The cone fishway design may therefore be particularly applicable to tropical rivers—like the Mekong—which have diverse communities of potamodromous species and are located in highly populated developing countries [6,11,16]. However, because of the newness of the cone fishway design, its effectiveness has so far only been initially evaluated in three Australian rivers (one tropical, one subtropical and one temperate), and these assessments focused on the effectiveness of the cone fishway in facilitating longitudinal passage of small-bodied diadromous species, since that was the initial intent of the fishway's design [15]. Consequently, cone fishway suitability for facilitating lateral migrations of potamodromous fishes in high biomass tropical river-floodplain systems, such as the Mekong, warrants empirical consideration.

The Mekong River has one of the most productive and diverse freshwater fisheries in the world [17] but is currently under immense pressure from widespread irrigation and hydropower development [2,5]. The 4800-km long river supports over 60 million people throughout six Asian countries (China, Myanmar, Lao PDR, Thailand, Cambodia, and Vietnam) [18], and is home to an estimated 2000 species of fish [17]. The fishery in the Lower Mekong Basin (LMB—the portion of the basin located in Lao PDR, Thailand, Cambodia and Vietnam) provides between 48% (for Laos) and 79% (for Cambodia) of animal protein intake, respectively [18]). Annual yield from the capture fishery forms approximately 2% of the world's total marine and freshwater catch [6]; consequently, the Mekong River fishery is crucial for supporting the food requirements and incomes of the neighboring human populations in the LMB [6].

Forty to seventy percent of fish catch in the Mekong consists of species that regularly undertake longitudinal and/or lateral migrations to access spawning, nursery, feeding or refuge habitat [19]. For instance, some of these species migrate from the main channel to floodplain wetlands and lakes during the wet season to access nursery habitat [6], while others live in floodplain wetlands and lakes during the wet season and migrate to tributaries or the main channel during the dry season [20]. Consequently, there is a critical need to develop and refine innovative fishway technologies to ameliorate the impacts of the proliferation of physical barriers in the LMB.

This study investigated the effectiveness of the new concrete cone fishway for facilitating lateral migrations of tropical river-floodplain fish communities in the LMB in Lao PDR. Assessments were conducted during both the day and dusk/night to consider the potential influence of differences in diurnal migration patterns, since previous studies have suggested that such differences may be an inherent aspect of the ecology of many tropical river-floodplain fish communities [7,11]. We hypothesised that the installation of the cone fishway would significantly enhance lateral fish migration from the Mekong River into an upstream wetland system during both the day and dusk/night.

## 2. Materials and Methods

### 2.1. Study Site

The Mekong River is one of the world's largest river systems, with a total drainage area of around 795,000 km$^2$ [21]. The LMB makes up approximately 78% of the total Mekong River basin area and has two main hydrological seasons—a wet season that typically lasts from June to October, and a dry season for the remaining months of the year [21]. Mean annual rainfall in the LMB ranges from more than 3000 mm in Lao PDR and Cambodia, to approximately 1000 mm in the semi-arid region of Northeast Thailand [22]. The Mekong River flows usually start increasing at the beginning of the wet season in May and peak in September/October at approximately 45,000 m$^3$s$^{-1}$. Discharges then decline again until they reach their minimum levels in March/April at approximately 1500 m$^3$s$^{-1}$ [23].

The cone fishway investigated in this study was located at the site of a floodplain regulator at Pak Peung village (103.696943° E, 18.348375° N; Bolikhamxay Province) in the LMB in central Laos (Figure 1). The regulator was 10 m high, with three sluice gates to control water transfers from Pak Peung wetland downstream via a flood runner into the Mekong River [6] (Figure 1). The regulator's sluice gates prevented the village's floodplain rice crops from being inadvertently flooded when wet season water levels rose and allowed for improved water security to support irrigation during the dry season. Prior to the installation of the regulator, fish could move upstream from the Mekong River to Pak Peung wetland via the flood-runner. However, the installation of the regulator resulted in upstream fish passage being totally obstructed, and fish only being able to move downstream through the sluice gates, from Pak Peung wetland into the Mekong River, when the gates were open [6]. Locals reported that many species disappeared from the wetland following regulator construction [24].

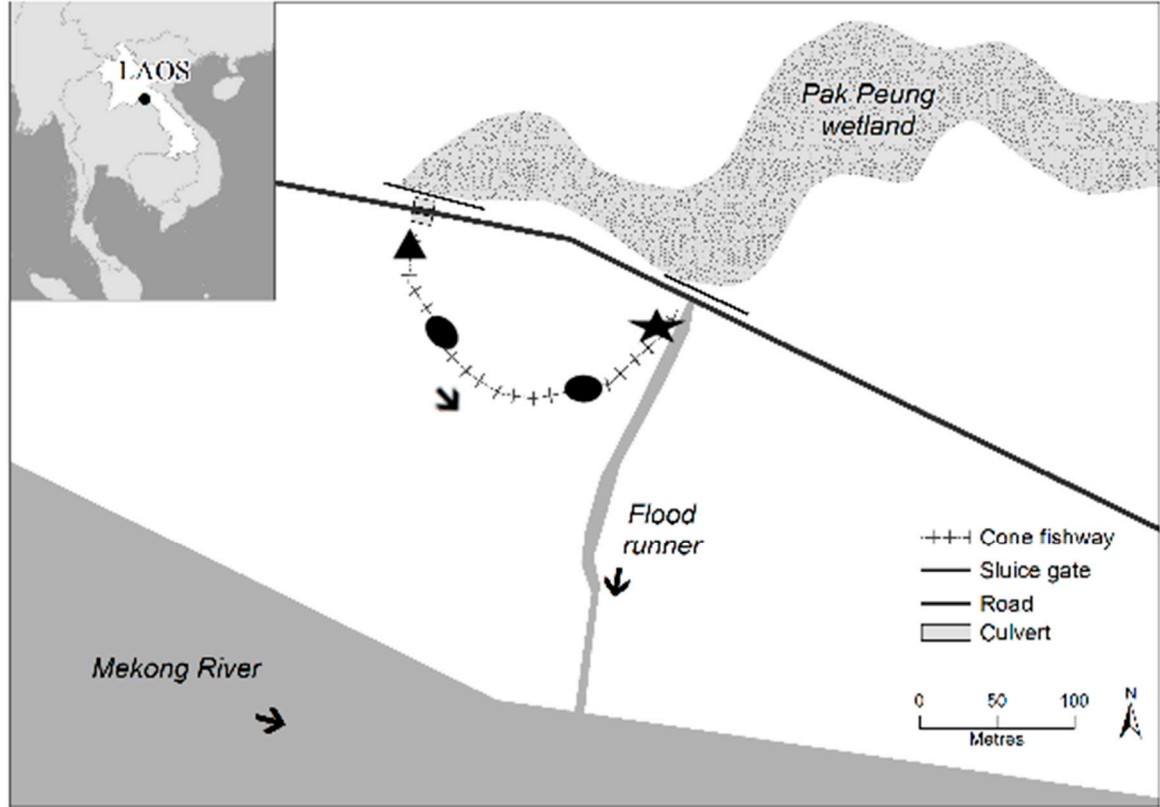

**Figure 1.** Site map of Pak Peung fishway (103.696943° E, 18.348375° N), in addition to a broader scale map of South East Asia displaying the location of the fishway within Lao PDR. Fish were sampled at the entrance (black star) and exit (black triangle) of the cone fishway (i.e., immediately downstream of the culvert). The arrows indicate the direction of flow in the fishway, flood runner and Mekong River. The black ovals on the fishway represent the long resting pools (source: Charles Sturt University GIS team using ARC GIS).

A cone fishway was installed at the Pak Peung regulator site during the dry season of 2012–13 to enable fish to move upstream to Pak Peung wetland. The fishway channel was 245 m long and contained 46 cone baffles. The baffles consisted of 1.0 m-high concrete cones and were arranged to have tapered 0.10–0.25 m wide slots between them (Figure 2). Two alternate baffle arrangements—baffles with three full slots and baffles with two full slots and two half slots—were applied to offset the slots and dissipate turbulence. Each set of baffles was separated by a 2 m long by 3 m wide pool with sloping sides and a flat bottom, which varied in depth between 0.6 and 1.5 m depending on the headwater level (Figure 2). There was a head differential of 0.09 m between pools, based on previous work which determined this to be optimal for Lower Mekong species [6]. The fishway had an average slot velocity of 1.33 m s$^{-1}$ and pool turbulence of 50 W m$^{-3}$ ($C_d = 0.6$) when it was operating within its intended slot depth range of 0.1–0.8 m [16]. This design arrangement was primarily targeted toward passage of fish <0.6 m long, but larger fish (e.g., whiprays up to 1 m long) have ascended similar cone fishways in tropical Australia [15].

To take advantage of the local topography, the fishway was divided into three sections, separated by two long pools (Figure 1). The lowest section contained 32 baffles, the middle section contained nine baffles, and the upper section contained five baffles. The long resting pool separating the lower and middle fishway sections was 45 m long, while the long pool separating the middle and upper fishway sections was 70 m long.

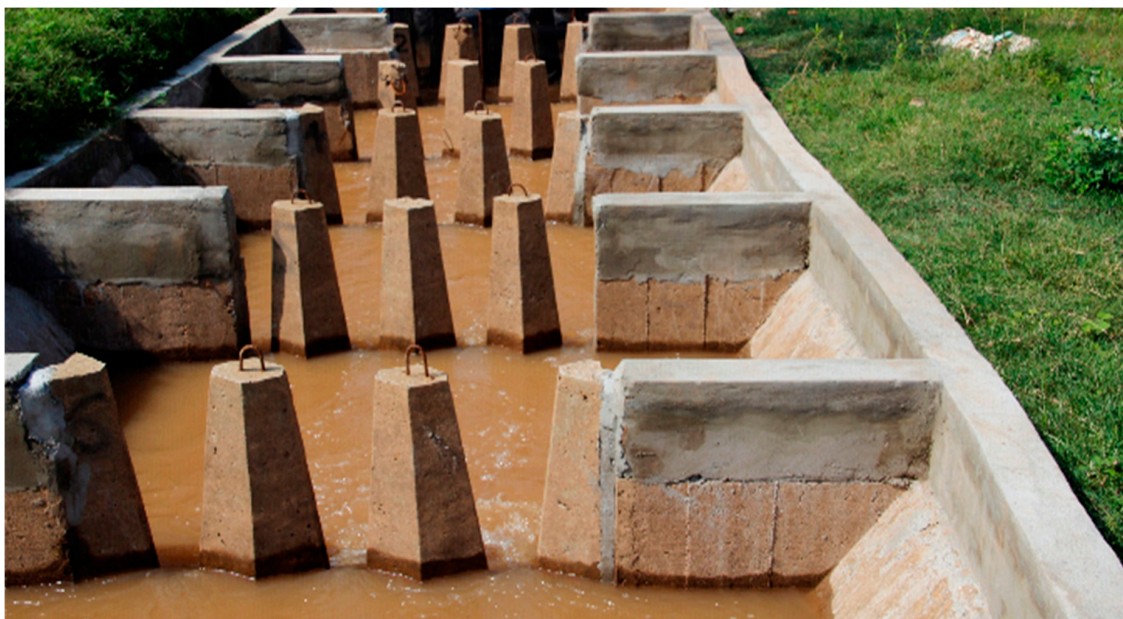

**Figure 2.** The cone fishway at Pak Peung village. The fishway had 1.0 m-high concrete cone baffles set up in two alternating arrangements—one with three full slots and the other with two full slots and two half slots (source: Lee Baumgartner). Each set of baffles was separated by a 2 m long by 3 m wide pool, which had a depth ranging from 0.6 to 1.5 m depending on the flow through the fishway. The sloping channel walls, low depth and low energy pools also made for safe human access and egress, which was a site-specific requirement in the remote village where children regularly interact with the fishway.

There was a 12 m long by 1.5 m deep concrete box culvert upstream of the upper fishway section, which traversed a road separating Pak Peung wetland from the Mekong River (Figure 1). A sluice gate regulator was constructed upstream of the culvert (i.e., at the culvert inlet) to regulate flows through the cone fishway and to prevent back-flooding into the wetland from the Mekong River during the wet season. The partial closure of the culvert sluice gate regulator during periods of high headwater levels created high current velocities immediately under the gate [16], but ensured that the cone fishway's internal hydraulics remained within their intended operating limits by maintaining water levels within the slot operating range of 0.1–0.8 m. The current study was specifically interested in quantifying the ability of fish to enter the cone fishway and ascend to the top of the fishway rather than the culvert, since any potential passage issues with the culvert were deemed to be context-specific and inapplicable to other sites.

## 2.2. Experimental Design

The effectiveness of the cone fishway was evaluated by undertaking two independent randomised block experiments—a day experiment investigating passage during diurnal periods, and a dusk/night experiment, investigating passage overnight and at crepuscular periods. For each experiment, we compared the species richness, abundance, biomass and size range (10th percentile, median and 90th percentile lengths) of fish that located and entered the fishway (i.e., reached the fishway entrance), with an independent sample of those that located, entered and passed the full length of the fishway (i.e., reached the fishway exit). This paired fishway entrance/exit randomised block design has been successfully applied in many other fishway evaluation studies [7,15,16], although very few studies have simultaneously considered the potential influence of diel variation in fish migration patterns [7]. We did not sample the upstream end of the culvert, as the focus of this study was on testing the passage effectiveness of the actual cone fishway and not the culvert (and thus we wished to remove the effect of the culvert; which is reported in [16]).

Both the day and dusk/night experiments were completed during the wet season of 2013 and involved 18 replicate blocks (where a 'block' was a partition in time). Within each block, each treatment for the day experiment ran for 7 h (between 8 a.m. and 3 p.m.); each block was completed over two consecutive days; and the order of the treatments within each block was randomised. The dusk/night experiment was performed using the same protocols as the day experiment, but each treatment ran for 16 h between 3.30 p.m. and 7.30 a.m. It was necessary to use a 'block' design in this study because the fish community changed each day as new fish approached the regulator seeking a migration route. Thus, when treatments were performed, there could have been a different population of fish every day.

Entrance samples were collected by placing a large fish trap (0.002 m mesh cone design, 1 m wide ×1 m tall ×2 m long) at the most downstream point of entry into the fishway. As the Mekong River levels can vary markedly, this location varied over the course of the experiment and was defined as the point within the fishway where tailwater depth had no influence on hydraulics to overcome the variable river levels. Exit samples were collected by placing the same fish trap at the most upstream location within the cone fishway. These methods were tailored to catch upstream migrating fish [16], and there were no observations of fish entering the fishway traps from other sources, such as downstream migrating fish from the wetland.

Before commencing an experimental block for each experiment, the fishway was flushed by implementing a high flow continuously for a period of 30 min which enabled us to apply a new set of hydraulic conditions to newly ascending fish [25]. The absence of fish was tested by netting each pool and confirming that the intervention had been successful. After each sampling period was completed, the trap was retrieved, and all fish were reassigned to a 60 L container with aerated river water for further processing. The trap was then reset and the remaining location treatment in the experimental block was sampled. All trapped fish were identified and weighed and a subset of 20 individuals per species were measured for length (TL).

### 2.3. Data Analysis

Prior to undertaking any analyses, all count data from the day and dusk/night experiments were converted to the standardised rate of fish trapped per hour of sampling. Randomised block design ANOVAs were performed separately for each experiment to examine the influence of the location in the fishway (i.e., entrance or exit) on the (1) average species richness (species per sample)); (2) average abundance (catch per unit effort (CPUE)); (3) average biomass (biomass per unit effort (BPUE)); (4) median, 10th percentile and 90th percentile lengths of all fish; and (5) mean lengths of the common species of fish trapped (common species were defined as the 10 most prevalent species—in terms of sample occurrence—recorded during the day and dusk/night experiments). Passage efficiency was also assessed by considering the total abundance of fish at the fishway exit (averaged across all species) as a percentage of the total abundance of fish at the fishway entrance (averaged across all species). The CPUE and BPUE datasets were initially square root $(x + 0.5)$ transformed to normalise their distributions and homogenise their variances, but the species richness and length (overall and common species) datasets were left untransformed because they were already homoscedastic and normally distributed. ANOVAs revealing significant main results were interrogated further by undertaking pairwise comparisons and Scheffe's correction for Type I errors [26].

The risk of sustaining any possible effects associated with the trap location treatment order in each experiment was alleviated by randomising the order of the trap location treatments for each block independently, making it reasonable to analyse the ANOVA error terms as independent [27]. Furthermore, the randomised block designs of both the day and dusk/night experiments removed the risk of temporal confounding between treatments for each experiment [11,16,28].

The fish biomass values were also used to estimate the biomass of fish potentially being passed by the cone fishway into Pak Peung wetland, using a fish biomass transfer estimation approach similar to that applied in Oldani and Baigun [29]. Specifically, the average day and dusk/night fish BPUE values (biomass.hour$^{-1}$) observed at the fishway exit were multiplied by their respective daily hours for the

fishway operating period during the 2013 wet season (which corresponded with the study's sampling period from 3 June to 13 July). The total day and dusk/night fish biomass transfer values were then summed to provide an estimate of the total fish biomass transfer for the 2013 fishway operating period.

Multivariate community analyses were undertaken separately for the day and dusk/night experiments to assess the influence of fishway location on the composition (the presence or absence of taxa) of fish communities passing through each fishway treatment, using presence-absence transformed CPUE (average catch per hour) data (sensu Baumgartner et al. [11]). The similarity matrices for the day and dusk/night experiment community composition data were calculated using the Jaccard Similarity measure. PERMANOVA (PERMANOVA+ for PRIMER) [30] was then used to examine the influence of fishway location. The similarity percentages procedure (SIMPER) was also undertaken to determine which taxa contributed to variation in community composition among treatments (i.e., fishway locations) for each experiment (SIMPER in Primer v6 [31]). Species were considered most important in discerning between treatments in SIMPER analyses, if they contributed at least 2% of total dissimilarity and their standard deviation ratio (Dissimilarity/Std.dev) was ≥1.

## 3. Results

### 3.1. Overall Results

An average CPUE of 54 fish.hour$^{-1}$ was observed in the fishway overall (i.e., for the entrance and exit samples in the day and dusk/night experiments). The total catch comprised of 100 species, which ranged in size from 22 to 370 mm TL (Table A1).

### 3.2. Fishway Entrance and Exit Patterns from the Day Experiment

Seventy-three taxa were sampled during the day (size range: 22–315 mm TL), and five of these taxa were exclusive to this experiment (Table A1). An average CPUE of 119 fish.hour$^{-1}$ (comprised of 67 taxa) was observed at the downstream entrance, while an average CPUE of 34 fish.hour$^{-1}$ (comprised of 58 taxa) was observed at the upstream exit. Fifty-two of the taxa occurred at both the entrance and exit, suggesting that at least 78% (52/67) of the entrance taxa were able to fully ascend the fishway. Passage efficiency averaged across all species was 29% (34 fish.hour$^{-1}$/119 fish.hour$^{-1}$) during the day.

ANOVA indicated that species richness and abundance were both significantly lower at the exit than that at the entrance (species richness: $F_{(1,17)} = 6.740$, $p = 0.0189$; abundance: $F_{(1,17)} = 10.600$, $p = 0.0047$) (Figure 3). Overall fish lengths at the exit were significantly longer than those at the entrance, and correspondingly, there was no difference in total fish biomass between the entrance and exit locations (10th percentile length: $F_{(1,17)} = 15.990$, $p = 0.0009$; median length: $F_{(1,17)} = 12.050$, $p = 0.0029$; biomass: $F_{(1,17)} = 1.540$, $p = 0.2312$) (Figure 3). Individuals of the common species, *Parambassis siamensis* ($F_{(1,17)} = 23.174$, $p = 0.0002$), *Puntioplites falcifer* ($F_{1,17} = 7.813$, $p = 0.0267$) and *Xenentodon* sp. ($F_{(1,17)} = 21.609$, $p = 0.0004$) in particular, were significantly longer at the exit than those at the entrance (Figure 4). PERMANOVA indicated that fish community (i.e., species) composition varied significantly between the entrance and exit locations (*Pseudo-F*$_{(1,17)} = 3.350$, $p = 0.0001$); and SIMPER revealed that *Rasbora daniconius*, *Puntius partipentazona* and *Parachela* spp. occurred in a lesser proportion of exit samples than entrance samples, whereas *Hypsibarbus lagleri*, *Amblyrhynchichthys micracanthus* and *Barbonymus altus* occurred in a greater proportion of exit samples than entrance samples (Table 1).

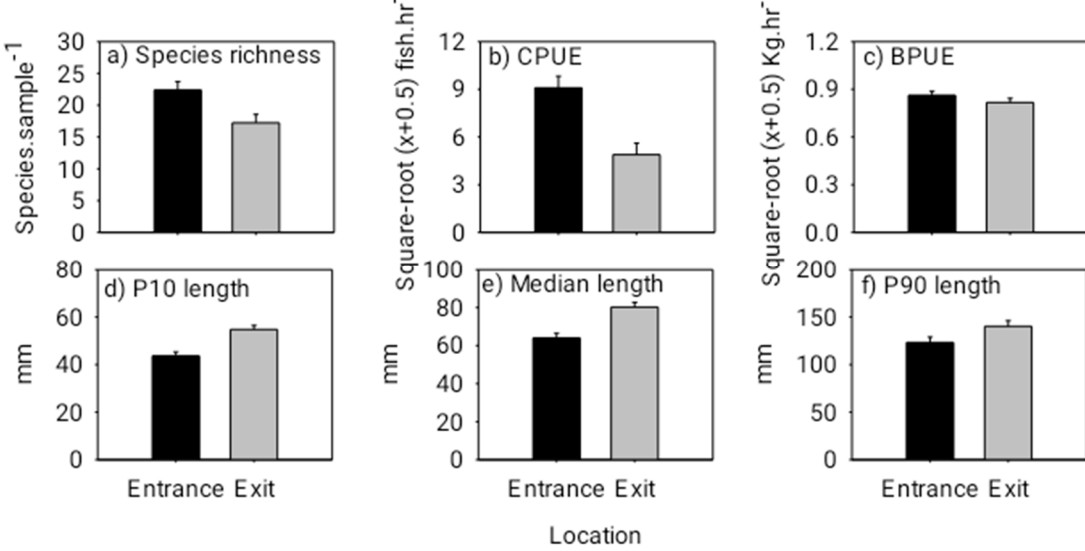

**Figure 3.** Mean (+ 1 SE) species richness (species sample$^{-1}$) (**a**), mean (+ 1 SE) abundance (catch per unit effort (CPUE) in catch hour$^{-1}$) (**b**), mean (+ 1 SE) biomass (biomass per unit effort (BPUE) in kg hour$^{-1}$) (**c**), 10th percentile (+ 1 SE) fish length (**d**), median (+ 1 SE) fish length (**e**) and 90th percentile (+ 1 SE) fish length (**f**) (lengths pooled across all species) at the entrance and exit locations for the day experiment. Note, the black bars represent the downstream fishway entrance and the grey bars represent the upstream exit. Additionally, square root (x + 0.5) transformed values have been presented for CPUE and BPUE to correspond with the ANOVA tests.

**Table 1.** Average prevalence (i.e., proportion of occurrence in samples) of species identified by SIMPER during the day experiment as important in contributing to between-treatment dissimilarity in community composition (based on presence-absence transformed community data). Each species contributed at least 2% of total dissimilarity and had a dissimilarity-to-standard deviation ratio of ≥1.

| Species | Entrance | Exit | Dissimilarity Contribution (%) |
|---|---|---|---|
| *Rasbora daniconius* | 0.72 | 0.17 | 3.04 |
| *Hypsibarbus lagleri* | 0.33 | 0.78 | 2.9 |
| *Amblyrhynchichthys micracanthus* | 0.33 | 0.72 | 2.84 |
| *Barbonymus altus* | 0.39 | 0.67 | 2.66 |
| *Puntius partipentazona* | 0.61 | 0 | 2.62 |
| *Parachela* spp. | 0.56 | 0.17 | 2.4 |

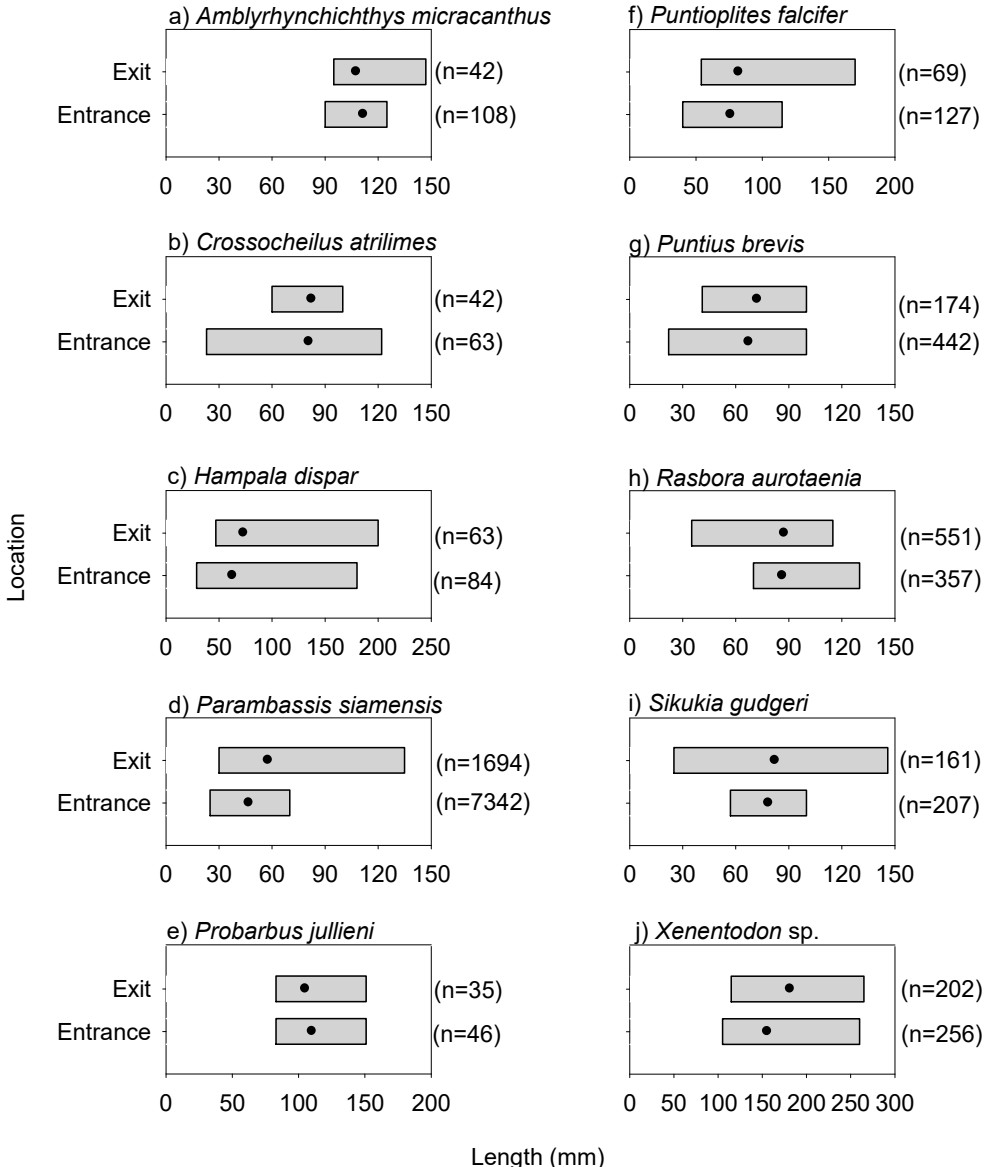

**Figure 4.** Length ranges of common fish species at the entrance and exit locations for the day experiment. Note, the solid black circles represent the mean values. Common species refers to the 10 most prevalent species (based on sample occurrence) recorded during both day and dusk/night experiments.

### 3.3. Fishway Entrance and Exit Patterns from the Dusk/Night Experiment

Ninety-five species were sampled during the dusk/night (size range: 25–370 mm TL), and 27 of these taxa were exclusive to this experiment (Table A1). An average CPUE of 38 fish.hour$^{-1}$ (comprised of 92 taxa) was observed at the entrance during the dusk/night, whereas an average CPUE of 21 fish.hour$^{-1}$ (comprised of 69 taxa) was observed at the exit. Sixty-six of the taxa occurred at both the entrance and exit, suggesting that at least 72% (66/92) of the entrance taxa were able to fully ascend the fishway. Passage efficiency averaged across all species was 55% (21 fish.hour$^{-1}$/38 fish.hour$^{-1}$) during the dusk/night.

The fishway locational patterns for the dusk/night experiment were similar to those for the day experiment. Specifically, species richness and abundance were again both significantly lower at the exit than that at the entrance during the dusk/night experiment (species richness: $F_{(1,17)} = 9.560$, $p = 0.0074$; abundance: $F_{(1,17)} = 14.220$, $p = 0.0019$) (Figure 5). Overall fish lengths at the exit were significantly longer than those at the entrance, and there was no difference in total fish biomass between the entrance and exit locations (10th percentile length: $F_{(1,17)} = 7.210$, $p = 0.0169$; median length: $F_{(1,17)} = 9.280$, $p = 0.0082$; biomass: $F_{(1,17)} = 0.140$, $p = 0.7161$) (Figure 5). Individuals of the common species, *Amblyrhynchichthys micracanthus* ($F_{(1,16)} = 7.990$, $p = 0.0198$) and *Parambassis siamensis* ($F_{(1,17)} = 11.146$, $p = 0.0053$), in particular, were significantly longer at the exit than those at the entrance (Figure 6). Additionally, fish community (i.e., species) composition again varied significantly between the entrance and exit locations (*Pseudo-F*$_{(1,15)} = 2.220$, $p = 0.0020$). This was largely due to *Poropuntius normani* and *Rasbora borapetensis* occurring in a lesser proportion of exit samples than entrance samples (Table 2).

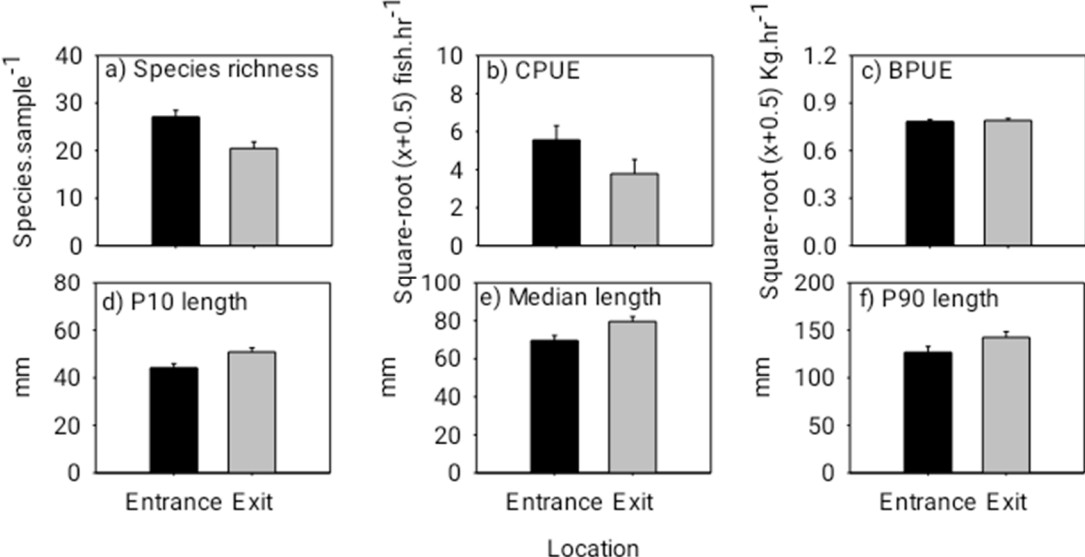

**Figure 5.** Mean (+ 1 SE) species richness (species sample$^{-1}$) (**a**), mean (+ 1 SE) abundance (CPUE in catch hour$^{-1}$) (**b**), mean (+ 1 SE) biomass (BPUE in kg hour$^{-1}$) (**c**), 10th percentile (+ 1 SE) fish length (**d**), median (+ 1 SE) fish length (**e**) and 90th percentile (+ 1 SE) fish length (**f**) (lengths pooled across all species) at the entrance and exit locations for the dusk/night experiment. Note, the black bars represent the entrance and the grey bars represent the exit. Additionally, square root (x + 0.5) transformed values have been presented for CPUE and BPUE to correspond with the ANOVA tests.

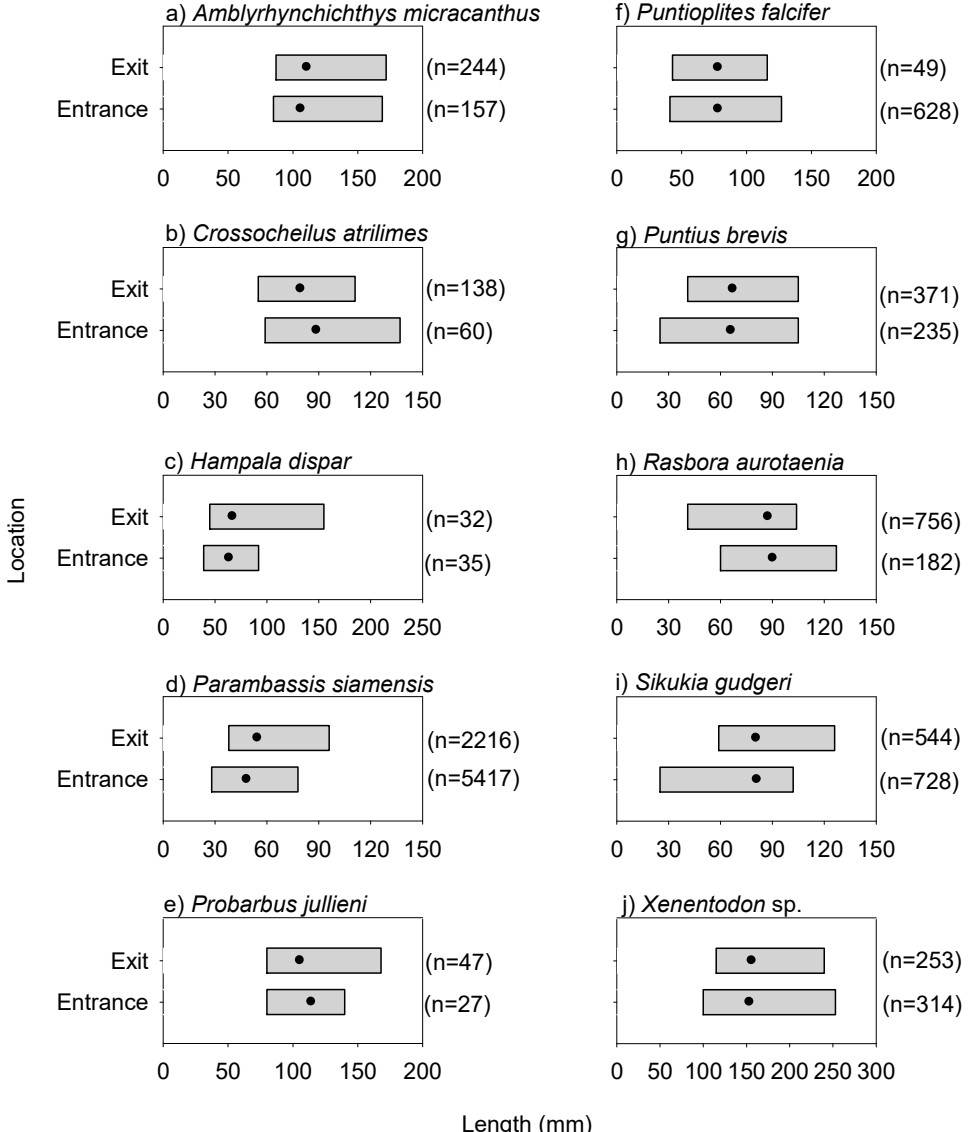

**Figure 6.** Length ranges of common species at the entrance and exit locations for the dusk/night experiment. Note, the solid black circles represent the mean values. Common species referred to the 10 most prevalent species (based on sample occurrence) recorded during both experiments.

**Table 2.** Average prevalence (i.e., proportion of occurrence in samples) of species identified by SIMPER during the dusk/night experiment as important in contributing to between-treatment dissimilarity in community composition (based on presence-absence transformed community data). Each species contributed at least 2% of total dissimilarity and had a dissimilarity-to-standard deviation ratio of ≥1.

| Species | Entrance | Exit | Dissimilarity Contribution (%) |
|---|---|---|---|
| *Poropuntius normani* | 0.65 | 0.18 | 2.44 |
| *Rasbora borapetensis* | 0.65 | 0.29 | 2.42 |

### 3.4. Fish Biomass Transfer facilitated by the Cone Fishway

An average biomass of 0.168 kg.h$^{-1}$ of fish passed the fishway exit during the day, whereas an average biomass of 0.124 kg.h$^{-1}$ of fish passed the same location during the dusk/night. Based on these BPUE values, it was estimated that the fishway facilitated the passage of 132.1 kg of fish from the Mekong River into Pak Peung wetland over the study period.

## 4. Discussion

The cone fishway facilitated the passage of the majority of species and size classes of fish that located and entered the fishway, during both the day and dusk/night experiments. Nevertheless, some small-bodied species and small individuals of large-bodied species (e.g., <50 mm long) appeared to be disproportionately limited in their ability to ascend the full length of the fishway. This suggests that the cone fishway is a relatively cost effective and low maintenance option for improving lateral fish passage in tropical river-floodplain systems like the Mekong, although its effectiveness for small fish could be further optimised.

### 4.1. Effectiveness of the Cone Fishway Design

For fish to successfully negotiate a fishway, they must be able to firstly, approach the fishway; secondly, locate and enter the fishway entrance; thirdly, ascend the full length of the fishway and fourthly, exit the fishway [7]. The current study involved undertaking day and dusk/night experiments to assess the second and third aspects—that is, the percentage of entrance-finding fish that could ascend the full length of the fishway (i.e., passage efficiency) [7,32]. While further work is needed to evaluate the efficiency of the fishway entrance in attracting migratory fish, high numbers of LMB fish, comprised of a diverse range of species and size classes were able to both locate and enter the bottom of the fishway, and fully ascend during both the day and dusk/night experiments (e.g., *Parambassis siamensis*, *Sikukia gudgeri* and *Xenentodon* sp.). Moreover, the overall species composition and overall size range of fish sampled at the exit of the cone fishway were similar to those in the neighbouring channel joining the Mekong River and floodplain wetland (L. Baumgartner unpub. data).

The cone fishway performance in the Mekong River is broadly consistent with that noted from initial evaluations of similar fishways in Australian tropical rivers [15]. We also noted a greater overall abundance of fish, and diversity of species and sizes than in temporary experimental vertical slot and submerged orifice fishway designs tested at Pak Peung in the preceding year, although this was likely due to the permanent nature of outflow from the cone fishway than any other inherent attraction advantage [11]. These results suggest that cone fishways have the potential to work effectively in supporting lateral fish migrations during both the day and dusk/night, if they are designed and operated appropriately for the site-specific conditions (which includes the presence of other crossing structures, like the culvert located upstream of the cone fishway in the present study).

Fish biomass transfer modelling supported the hypothesis that the cone fishway would enhance lateral fish migration during both the day and dusk/night. Furthermore, biomass transfer did not appear to be reduced by the fishway, since there was no difference in the total biomass of fish at the entrance and exit, for both the day and dusk/night. Hence, while some small fish (<50 mm long) had reduced passage, our results suggest that the integrity of river-floodplain biomass exchange was maintained. In addition, a number of small black fish species (i.e., species that typically only live in floodplain lakes and wetlands) ascending the fishway were observed to be ovigerous (i.e., carrying eggs), suggesting that these fish were migrating to spawn in the wetland, likely further augmenting productivity [33]. Indeed, the relatively moderate biomass of fish transferred to Pak Peung wetland may result in greater fish biomass within the wetland over the course of several years, via spawning, growth and recruitment [33]. According to Ferguson et al. [20], some black fishes move to refuge pools in nearby rivers in the dry season, and then back to their preferred floodplain habitats for the rest of the year. While the reproductive status of the black fish in the present study provides for an area for further study, these fish were likely moving back into the wetland after being displaced by high flows. This observation further reinforces the value of effective fishways in facilitating access for fish to access their preferred habitats for a variety of obligatory lifecycle processes. If the fishway was not present, these fish would have been permanently displaced into unsuitable habitat.

The success of the cone fishway at Pak Peung was likely related to its relatively conservative internal hydraulics. In particular, the cone fishway had a lower average pool turbulence (i.e., energy

dissipation factor) of 50 W m$^{-3}$ (C$_d$ = 0.6) and water velocity (maximum 1.33 m s$^{-1}$) than prior fishways in the region [11,16]. This was a key, and deliberate, design consideration.

Yet, despite the overall success of the cone design and passage of fish as small as 50 mm TL, there were fewer species and individuals at the exit than at the entrance, and the fish at the exit were significantly larger than those at the entrance. The entrance and exit fish communities also had differing species compositions, suggesting that not all fish could ascend the full length of the cone fishway—especially the smaller (<50 mm TL) species (e.g., *Akysis ephippifer*) and younger life-stages of larger-bodied species. Other studies have similarly observed significantly smaller fish at the entrance of cone fishways [15,34] but reported that the cone fishways nonetheless facilitated passage for an extensive size range of small-bodied fish overall. Indeed, Stuart and Marsden (2019) evaluated the effectiveness of cone fishways in the Norman, Fitzroy and Maribyrnong rivers (Australia), where a broad suite of the local fish community, including fish as small as 9 mm long, successfully passed.

The smaller fish in the current study may have been constrained by the water velocity and/or turbulence levels, since these fish have lesser absolute burst swimming abilities than large fish [28,35]. The reduced passage success of small fish may have also been related to the overall length of the fishway. The Pak Peung regulator was 10 m high, resulting in a 245 m long fishway, which in this case may have exceeded the energetic resources of some of the local fish [36]. Nevertheless, these were key design trade-offs that were considered prior to constructing the fishway. To shorten the fishway whilst maintaining low turbulence would have required deeper pools, but this would have had cost-implications beyond what the project budget could accommodate. Consequently, two larger resting pools were included.

In addition to the two large resting pools within the Pak Peung cone fishway, a range of other potential solutions are available to potentially improve passage of small fish. These include providing larger cone pools, reducing overall discharge and retrofitting dissipaters [37]. In some cases, however, there are ecological trade-offs associated with such modifications. Reducing fishway discharge also reduces fish attraction, and in turn, passage rates. Such compromises reflect the challenges associated with designing fishways to meet a broad variety of migration needs among species, life-stages, species, flow settings and/or seasons [38]. Where barrier removal is unfeasible then transparent fish passage performance standards need to be developed, depending on the restoration objectives, available budget, site environmental conditions (i.e., topography, flow regime etc.) and knowledge of the migratory species [12]. For example, it may be possible to improve the suitability of the cone fishway design for small-bodied LMB fish by applying more conservative design criteria such as a reduced slope, decreased discharge, or greater pool size [12], although such modifications increase fishway length and cost, and may adversely influence the efficacy for large-bodied species. Future research should attempt to assess the impacts of these modifications on the fishway's entrance attraction efficiency and the migratory community in general (biomass, species composition, size structure)—each within the context of the broader downstream community [15].

### 4.2. Diurnal Patterns

At Pak Peung, total fish abundance, species richness and biomass were greater during the day than during dusk/night (although these diurnal differences varied greatly among individual species). In other parts of Laos and in Australia, there can also be higher abundances of fish during the day [11,39]. At Pak Peung, some fish may have avoided using the fishway at night as has been reported for fishways in the lower Río de la Plata basin (Argentina–Paraguay) [29]. Diurnal changes in fish migration rates are common [40], but have not received much attention in the LMB (although see Baumgartner et al. [11]). In long fishways, such as at Pak Peung, fish with strong diurnal movement preferences need to be able to complete their ascent during their preferred diurnal period, and further research is needed to clarify whether this is a limiting criterion. Similarly to that observed for individual species in other paired fishway entrance/exit studies [7], several species (*Hypsibarbus lagleri*, *Amblyrhynchichthys micracanthus* and *Barbonymus altus*) occurred in a greater proportion of exit samples than entrance samples during

our day experiment, possibly because these species aggregated in the resting pools before continuing their ascent, and subsequently exited in greater numbers than they entered at the bottom. These patterns could potentially be tested in future studies by following the movements of individual fish throughout the fishway using a mark-recapture approach such as passive integrated transponder (PIT) tagging [41].

### 4.3. Management Implications and Areas for Future Research

Ecological targets for potamodromous fishes have advanced from passing adults of a few species to passing whole fish communities comprised with a diverse range of movement strategies, including small and large fish [28]. The cone fishway—whilst having a limited headwater operating range—offers a viable option for supporting passage of a diverse range of fish species and size classes at low/medium-head (i.e., <10 m high) weirs on tropical rivers. The cone fishway has the inherent advantages of (1) a relatively simple, low maintenance design and a low overall build cost; (2) the potential for low turbulence pools; (3) an ability to conserve water (e.g., to prevent the headwater from being drained); and (4) offering the potential for safer human access and egress, which is a high priority in developing countries where regional communities regularly operate/access riverine infrastructure.

The optimal solution to restoring fish passage at any site is to remove the physical barrier causing the issue. However, given that this is not possible at Pak Peung, the cone fishway provides an excellent alternative, and has the potential to support a marked increase in wetland fish productivity during the period of the fishway's life time (40–50 years). This is a far better outcome than having no fishway, and thus no fish movement between the river and the wetland.

Further context-specific field assessments of new cone fishways will be crucial to optimise future applications, refine design features and optimize entrance attraction, passage efficiency and exchange of biota between rivers and floodplains. There is also a need to ensure that cone fishways can adequately pass the high fish biomasses often found in tropical rivers. Fish passage is an adaptive science. Lessons are learned at new sites and applied to improve future projects. The information collected at Pak Peung will serve as examples of how to achieve better fish passage at other sites where irrigation structures are expected to impact fisheries productivity. Migratory river fishes comprise a very large group of species which are now at unprecedented risk from infrastructure development [42]; effective fishways are needed to help preserve ecosystem integrity of large tropical systems globally.

**Author Contributions:** L.J.B. conceived the project idea and was responsible for funding acquisition; L.J.B., C.B., T.M., J.M., O.P., D.S., I.G.S., and G.T. designed the study and collected field data; W.R. and N.N. analysed the data; L.J.B. and N.N. wrote the manuscript; and all authors contributed critically to the drafts and approved the final manuscript version. All authors have read and agreed to the published version of the manuscript.

**Funding:** This research was funded by Primary Industries New South Wales, Charles Sturt University and the Australian Centre for International Agricultural Research (project FIS/2009/041).

**Acknowledgments:** We thank all of the people and villagers of Pak Peung who assisted with this study. We are especially grateful to Phannonlath, Sanniravong, Vone and Anousai for their support. Kate Martin, Andrew Trappett and Matthew Barwick assisted with fieldwork. Martin Mallen-Cooper and Chris Barlow had high-level roles in the design of the study.

**Conflicts of Interest:** The authors declare no conflict of interest. The funders had a high-level role in the design of the study; but no hands on role in the collection, analyses, or interpretation of data; in the writing of the manuscript, or in the decision to publish the results.

# Appendix A

**Table A1.** Summary of fish species caught at the cone fishway entrance and exit, and their respective length ranges for the day and dusk/night experiments (mm TL).

| Species | Total N | Day Experiment | | Dusk/Night Experiment | |
|---|---|---|---|---|---|
| | | **Entrance** | **Exit** | **Entrance** | **Exit** |
| *Acanthopsis* spp. | 7 | 140–155 | 149–149 | 121–152 | 138–138 |
| *Akysis ephippifer* | 8 | | | 27–42 | |
| *Amblyrhynchichthys micracanthus* | 551 | 90–125 | 95–147 | 85–169 | 87–172 |
| *Amblyrhynchichthys truncatus* | 2 | | | 107–122 | |
| *Anabas testudineus* | 203 | 40–95 | 155–162 | 40–63 | 119–119 |
| *Badis ruber* | 5 | 39–45 | | 35–41 | |
| *Barbonymus altus* | 199 | 25–102 | 40–86 | 35–95 | 40–103 |
| *Channa gachua* | 1 | | | 174–174 | |
| *Channa striata* | 57 | 25–285 | 91–91 | 69–250 | |
| *Chela laubuca* | 1 | | | | 80–80 |
| *Chitala blanci* | 1 | | | 100–100 | |
| *Chitala ornata* | 23 | | | 83–140 | 110–157 |
| *Cirrhinus cirrhosus* | 12 | 80–111 | 71–71 | 55–120 | 50–50 |
| *Clarias macrocephalus* | 3 | | | 39–106 | 112–112 |
| *Clarias* sp. (cf. *batrachus*) | 2 | | | 134–134 | 105–105 |
| *Clupeichthys aesarnensis* | 3456 | 34–64 | 52–65 | 35–69 | 40–54 |
| *Crossocheilus atrilimes* | 303 | 23–122 | 60–100 | 59–137 | 55–111 |
| *Cyclocheilichthys apogon* | 3 | 69-69 | | 80–105 | |
| *Cyclocheilichthys armatus* | 22 | 70–115 | 106–111 | 67–100 | 70–111 |
| *Cyclocheilichthys enoplos* | 28 | 81–144 | 92–154 | 98–111 | 95–117 |
| *Cyclocheilichthys lagleri* | 34 | 77–115 | 90–130 | 42–114 | 65–120 |
| *Cyprinus carpio* | 9 | 80–119 | 100–100 | 116–139 | 98–98 |
| *Esomus metallicus* | 409 | 45–82 | | 31–85 | 42–82 |
| *Hampala dispar* | 214 | 29–180 | 47–200 | 39–92 | 45–155 |
| *Hampala macrolepidota* | 26 | 43–73 | 54–168 | 65–83 | 57–82 |
| *Hemibagrus nemurus* | 8 | | | 195–220 | 165–230 |
| *Hemibagrus* spp. | 20 | | | 56–155 | 194–240 |
| *Henicorhynchus lobatus* | 96 | 60–133 | 65–170 | 54–135 | 55–145 |
| *Henicorhynchus ornatipinnis* | 30 | 80–119 | 78–102 | 71–98 | 65–96 |
| *Henicorhynchus siamensis* | 449 | 31–95 | 40–185 | 49–112 | 55–145 |
| *Homaloptera smithi* | 193 | 36–36 | 33–33 | 29–40 | 30–41 |
| *Hypophthalmichthys nobilis* | 1 | | 92–92 | | |
| *Hypsibarbus lagleri* | 170 | 36–111 | 43–125 | 37–105 | 46–163 |
| *Hypsibarbus malcolmi* | 63 | 80–128 | 77–141 | 75–135 | 79–127 |
| *Hypsibarbus wetmorei* | 14 | 67–90 | | 90–130 | 105–150 |
| *Kryptopterus bicirrhis* | 1 | | | 152–152 | |
| *Kryptopterus cryptopterus* | 2 | | | 145–145 | 175–175 |
| *Labiobarbus leptocheilus* | 709 | 90–162 | 83–195 | 69–160 | 51–168 |
| *Labiobarbus siamensis* | 8 | | 99–115 | 95–130 | |
| *Laides longibarbis* | 1 | | | 90–90 | |
| *Macrognathus semiocellatus* | 63 | 145–221 | 145–175 | 62–225 | 130–155 |
| *Macrognathus siamensis* | 57 | 80–231 | 170–202 | 109–198 | 152–190 |
| *Mastacembelus armatus* | 12 | 135–215 | | 134–205 | 230–230 |
| *Mastacembelus favus* | 6 | 120-315 | | 140-300 | |
| *Monopterus albus* | 3 | | | 63–109 | 220–220 |
| *Mystacoleucus ectypus* | 7 | 45–65 | 68–68 | | 57–57 |
| *Mystacoleucus marginatus* | 29 | 43–70 | 52–70 | 45–69 | 57–62 |
| *Mystus albolineatus* | 6 | 187–187 | | 161–250 | 125–216 |
| *Mystus atrifasciatus* | 40 | 52–85 | 63–89 | 45–78 | 40–180 |
| *Mystus mysticetus* | 2 | | 115–115 | 122–122 | |

**Table A1.** *Cont.*

| Species | Total N | Day Experiment | | Dusk/Night Experiment | |
|---|---|---|---|---|---|
| | | **Entrance** | **Exit** | **Entrance** | **Exit** |
| *Mystus singaringan* | 2 | | | 175-200 | |
| *Nandus oxyrhynchus* | 16 | 60–92 | 80–80 | 50–78 | 63–82 |
| *Nemacheilus longistriatus* | 109 | 55–82 | 61–78 | 57–68 | 59–75 |
| *Nemacheilus pallidus* | 2 | 59–76 | | | |
| *Neodontobutis aurarmus* | 3 | | | 37–37 | 29–30 |
| *Notopterus* | 36 | | 280–280 | 281–281 | 193–370 |
| *Ompok bimaculatus* | 15 | | | 65–91 | |
| *Oreochromis niloticus* | 21 | 51–313 | 265–265 | 85–173 | 81–285 |
| *Osteochilus hasselti* | 12 | 61–170 | 78–78 | | 134–191 |
| *Osteochilus lini* | 27 | 45–95 | 94–120 | 36–155 | 60–84 |
| *Osteochilus waandersii* | 1 | | 134–134 | | |
| *Oxyeleotris marmorata* | 10 | | | 30–55 | |
| *Pangasius macronema* | 6 | | | 77–97 | 85–90 |
| *Parachela siamensis* | 326 | 42–135 | 91–142 | 43–125 | 89–136 |
| *Parachela* spp. | 2024 | 35–95 | 45–68 | 40–75 | 35–76 |
| *Parambassis siamensis* | 16669 | 25–70 | 30–135 | 28–78 | 38–96 |
| *Parasikukia maculata* | 13 | 63–63 | | 48–65 | |
| *Poropuntius normani* | 79 | 35–71 | 25–210 | 37–90 | 48–76 |
| *Pristolepis fasciata* | 9 | 121–121 | 77–77 | 50–145 | 85–102 |
| *Probarbus jullieni* | 155 | 83–151 | 83–151 | 80–140 | 80–168 |
| *Pseudolais pleurotaenia* | 4 | | | 115–115 | 86–105 |
| *Puntioplites falcifer* | 873 | 40–115 | 54–170 | 41–127 | 43–116 |
| *Puntius aurotaeniatus* | 6 | | | 31–55 | |
| *Puntius brevis* | 1222 | 22–100 | 41–100 | 25–105 | 41–105 |
| *Puntius orphoides* | 7 | | 75-80 | 78–78 | 70–150 |
| *Puntius partipentazona* | 263 | 31–54 | | 30–52 | 38–46 |
| *Puntius proctozystron* | 170 | 38–106 | 56–132 | 47–115 | 50–110 |
| *Puntius stolickzcanus* | 1 | 45–45 | | | |
| *Raiamas guttatus* | 39 | 62–135 | 105–128 | 55–168 | 71–126 |
| *Rasbora aurotaenia* | 1846 | 70–130 | 35–115 | 60–127 | 41–104 |
| *Rasbora borapetensis* | 286 | 34–59 | 38–60 | 33–61 | 38–58 |
| *Rasbora daniconius* | 1153 | 37–90 | 58–95 | 51–98 | 51–90 |
| *Rasbora dusonensis* | 5 | | | 55–65 | |
| *Rasbora trilineata* | 411 | 43–87 | 50–78 | 33–74 | 48–75 |
| *Rhinogobius mekongianus* | 1 | | | 68–68 | |
| *Scaphognathops stejnegeri* | 43 | 52-95 | 56–86 | 25–81 | 60–80 |
| *Sikukia gudgeri* | 1640 | 57–100 | 25–146 | 25–102 | 59–126 |
| *Tenualosa thibaudeaui* | 8 | 113–113 | 85–145 | 74–100 | |
| *Thynnichthys thynnoides* | 70 | 80–170 | 125–182 | 140–240 | 92–172 |
| *Toxotes chatareus* | 11 | 82–155 | 95–95 | 86–113 | |
| *Trichopodus microlepis* | 7 | 56–78 | | 111–134 | 102–102 |
| *Trichopodus pectoralis* | 2 | 60–60 | | 185–185 | |
| *Trichopodus trichopterus* | 7 | 49–95 | | 75–75 | 91–91 |
| *Trichopsis vittata* | 1 | | | 61–61 | |
| Unknown 1 | 1 | | | 99–99 | |
| Unknown 2 | 1 | 90–90 | | | |
| *Wallago attu* | 2 | | | 144–147 | |
| *Xenentodon* sp. | 1025 | 105–260 | 115–265 | 100–253 | 115–240 |
| *Yasuhikotakia lecontei* | 30 | 45–66 | 45–66 | 46–62 | 50–52 |
| *Yasuhikotakia morleti* | 1 | | | 50–50 | |

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
