# Peer review of "A Cone Fishway Facilitates Lateral Migrations of Tropical River-Floodplain Fish Communities"

_water, doi:10.3390/w12020513_

Round 1

Reviewer 1 Report

Comment on Baumgartner et al.

This study examined the effectiveness of a new cone fishway for facilitating lateral migrations of river floodplain fish communities in the Lower Mekong Basin. This study is timely and important and is likely of great interest to managers and stakeholders in order to protect the biodiversity and the productivity of freshwater fisheries over the world.

On Figure 1, it would be nice to show where the entrance/exit of the cone are and where the sampling was carried out.

Line 211. Please add a reference for the correction applied.

In the abstract, it is stated that “the cone fishway design has great potential for facilitating the lateral migrations of diverse tropical river-floodplain fish communities at low/medium head infrastructure”. However, this does not seem to be the case according the results of the experiment where only 30% of the individuals have been able to fully ascend the fishway. Species richness was also lower at the exit. I would thus be more cautious about the efficiency of this design to facilitate lateral migrations. Furthermore, the fact that abundance is significantly lower at the exit than at the entrance but that biomass is similar at both extremes of the fishway suggests that small individuals do not succeed in the fishway. This is confirmed by the fact that fish length is higher at the exit than at the entrance on average. This is problematic given that floodplains are usually used as nurseries… Thus, the ideal design should allow adults to enter the floodplain to spawn but also juveniles (and adults) to exit the floodplain and go back to the river. If the juveniles are systematically killed, this may have long-term consequences on populations (both in term of size structure but also abundances) and species that you cannot see with this short-term experiment. Maybe worth discussing this further?

Perhaps it’s a silly question but is it feasible for the fish to migrate laterally without the cone fishway? My guess is yes because the aim of the fishway is to “facilitate” lateral migration, not to “allow” lateral migration. It would thus have been interesting to also estimate the migration pattern without the cone fishway and to compare it with the cone fishway. This estimate would have act as a reference (control) against which the efficiency of the cone fishway would have been compared. Without this control, it is really hard to appreciate the efficiency of the cone fishway. For instance, line 342, it is said that “the fishway facilitated the passage of 132.1kg of fish” but maybe a similar amount would have pass without the fishway. If I’m right, then it would be nice to discuss this limit of the study.

Line 430. Delete “in” (repeated word).

Author Response

Reviewer 1

This study examined the effectiveness of a new cone fishway for facilitating lateral migrations of river floodplain fish communities in the Lower Mekong Basin. This study is timely and important and is likely of great interest to managers and stakeholders in order to protect the biodiversity and the productivity of freshwater fisheries over the world.

1.On Figure 1, it would be nice to show where the entrance/exit of the cone are and where the sampling was carried out.

Authors’ response

We have amended the figure as suggested to show the location of the fishway entrance and exit (and thus where the sampling was carried out) using symbols. We have also updated the caption to explain that the triangle depicts the exit and that the star depicts the entrance. Specifically, the sentence in the caption ‘Fish were sampled at the entrance and exit of the cone fishway (i.e. immediately downstream of the culvert).’ has been replaced with ‘Fish were sampled at the entrance (black star) and exit (black triangle) of the cone fishway (i.e. immediately downstream of the culvert).’  Thankyou for the chance to clarify this aspect.

2.Line 211. Please add a reference for the correction applied.

Authors’ response

We have added the relevant reference: Scheffe, H. A method for judging all contrasts in the analysis of variance. Biometrika 1953, 40, 87-110. We have also updated the reference numbers accordingly.

3.In the abstract, it is stated that “the cone fishway design has great potential for facilitating the lateral migrations of diverse tropical river-floodplain fish communities at low/medium head infrastructure”. However, this does not seem to be the case according the results of the experiment where only 30% of the individuals have been able to fully ascend the fishway. Species richness was also lower at the exit. I would thus be more cautious about the efficiency of this design to facilitate lateral migrations. Furthermore, the fact that abundance is significantly lower at the exit than at the entrance but that biomass is similar at both extremes of the fishway suggests that small individuals do not succeed in the fishway. This is confirmed by the fact that fish length is higher at the exit than at the entrance on average. This is problematic given that floodplains are usually used as nurseries… Thus, the ideal design should allow adults to enter the floodplain to spawn but also juveniles (and adults) to exit the floodplain and go back to the river. If the juveniles are systematically killed, this may have long-term consequences on populations (both in term of size structure but also abundances) and species that you cannot see with this short-term experiment. Maybe worth discussing this further?

Authors’ response

We agree that by stating that the cone fishway has ‘great potential’ slightly overstates our results. We have modified both the Abstract and Discussion to be more conservative in this aspect. In the Abstract, we have moderated our language as follows. We have replaced:

‘These results suggest that the cone fishway design has great potential for facilitating the lateral migrations of diverse tropical river-floodplain fish communities at low/medium head infrastructure.

with:

‘These results suggest that, with further development, the cone fishway design has considerable potential for facilitating the lateral migrations of diverse tropical river-floodplain fish communities at low/medium head infrastructure.

4.Perhaps it’s a silly question but is it feasible for the fish to migrate laterally without the cone fishway? My guess is yes because the aim of the fishway is to “facilitate” lateral migration, not to “allow” lateral migration. It would thus have been interesting to also estimate the migration pattern without the cone fishway and to compare it with the cone fishway. This estimate would have act as a reference (control) against which the efficiency of the cone fishway would have been compared. Without this control, it is really hard to appreciate the efficiency of the cone fishway. For instance, line 342, it is said that “the fishway facilitated the passage of 132.1kg of fish” but maybe a similar amount would have pass without the fishway. If I’m right, then it would be nice to discuss this limit of the study.

Authors’ response

Your question is reasonable, but to clarify, it is entirely impossible for fish to migrate laterally (i.e. from the Mekong into the wetland) as they cannot negotiate the high (> 5m) wetland regulating barrier. An exception to this is if there is a major overbank flood but these are relatively rare (1:100 years). The fishway was a pilot site and the first attempt in South East Asia to provide the mechanism to improve this connectivity across a very broad range of flow conditions. We have clarified the first paragraph on the Introduction to indicate that fish are ‘prevented’ rather than ‘hampered’ and we have deleted the term ‘allowed’ in favour of ‘facilitated’. We now consider it clear that the fishway ‘facilitated’ passage and there were no other options that ‘allowed’ fish to migrate laterally into the wetland. Thankyou for the chance to clarify this aspect.

Please also see our response to comment 14 by Reviewer 3.

5.Line 430. Delete “in” (repeated word).

Authors’ response

This typo has now been deleted. Thanks for picking this up.

Reviewer 2 Report

Many fish species undertake more or less extended migrations as part of their behavior. River barriers often have a negative impact on natural fish populations and, with other factors, can contribute to the reduction of numbers, disappearance and even extinction of species. Fish passes are offered more often than ever before and new projects are needed to provide passage for entire fish communities. Despite recent progress, fish passes have performed poorly, especially for fish with a small body. Why some fishways work better than others and why some species perform better than others in fish passes is poorly understood. Therefore, the results of such studies are of great importance for fish assemblages, and in the case of a system such as the lower Mekong also for the local community. The reviewed manuscript is very well written, the text is understandable and the figures clearly show the results of the experiments. This is a valuable document presenting the results of assessing the effectiveness of a cone fishway. Hence, I believe that this manuscript is well timed and appropriate. In general, I have only a few minor comments.

Line 192: Please add some details about fish flushing in the fishway. Have you checked that all fish have been flushed?

Line 200: I’m not sure if randomised block design ANOVA can be applied to such an experimental design. Only two variables were analyzed.

Author Response

Reviewer 2

Many fish species undertake more or less extended migrations as part of their behaviour. River barriers often have a negative impact on natural fish populations and, with other factors, can contribute to the reduction of numbers, disappearance and even extinction of species. Fish passes are offered more often than ever before and new projects are needed to provide passage for entire fish communities. Despite recent progress, fish passes have performed poorly, especially for fish with a small body. Why some fishways work better than others and why some species perform better than others in fish passes is poorly understood. Therefore, the results of such studies are of great importance for fish assemblages, and in the case of a system such as the lower Mekong also for the local community. The reviewed manuscript is very well written, the text is understandable and the figures clearly show the results of the experiments. This is a valuable document presenting the results of assessing the effectiveness of a cone fishway. Hence, I believe that this manuscript is well timed and appropriate. In general, I have only a few minor comments.

1.Line 192: Please add some details about fish flushing in the fishway. Have you checked that all fish have been flushed?

Authors’ response

Thanks for pointing this out. We have modified the sentence from: “Before commencing an experimental block for each experiment, the fishway was flushed of all fish by implementing a high flow continuously for a period of 30 minutes.”

to:

“Before commencing an experimental block for each experiment, the fishway was flushed by implementing a high flow continuously for a period of 30 minutes which enabled us to apply a new set of hydraulic conditions to newly ascending fish (Mallen-Cooper 1999). The absence of fish was tested by netting each pool and confirming that the intervention had been successful.’

We could not fully confirm all fish left such a long fishway but our observations were that a period of flushing pushed fish back downstream and enabled us to begin the next treatment with the desired set of hydraulic conditions.

Such a method has been used commonly in previous studies:

Mallen-Cooper, M., 1999. Developing fishways for non-salmonid fishes; a case study from the Murray River in Australia. Innovations in Fish Passage Technology’. (Ed. M. Odeh.) pp, pp.173-195. Stuart, I.G. and Berghuis, A.P., 2002. Upstream passage of fish through a vertical‐slot fishway in an Australian subtropical river. Fisheries Management and Ecology, 9(2), pp.111-122.

Please also see our response to comment 16 by Reviewer 3.

2.Line 200: I’m not sure if randomised block design ANOVA can be applied to such an experimental design. Only two variables were analysed.

Authors’ response

Thankyou for the chance to clarify this aspect – we can see how it could be confusing. Perhaps this confusion is created by the reviewer only perceiving the term ‘block’ in a spatial context (for example, a series of blocks of land randomly divided into say two different agricultural treatments). However, we used temporal rather than spatial blocks and applied different fishway treatments in a randomised order within a two day ‘temporal’ block of treatments. In essence, the design we used is a crossover between a traditional repeated measures design (temporally applied treatments) and a traditional randomised complete block design (spatially applied treatments). The analysis is identical for either method, and is termed a randomised block design analysis. We hope this clarifies our approach, and have made the following changes to the Methods to better explain the actual randomised block design that we used in the paper. Specifically, we have replaced:

‘Both experiments were completed during the wet season of 2013 and involved eighteen replicate trials (experimental blocks). Each replicate trial for the day experiment ran for seven hours (between 8 am and 3 pm), and each block was completed over two consecutive days. The dusk/night experiment was performed using the same protocols as the day experiment, but each replicate trial ran for 16 hours between 3.30 pm and 7.30 am.’

with:

‘Both experiments were completed during the wet season of 2013 and involved eighteen replicate blocks (where a ‘block’ was a partition in time). Within each block, each treatment for the day experiment ran for seven hours (between 8 am and 3 pm); each block was completed over two consecutive days; and the order of the treatments within each block was randomized. The dusk/night experiment was performed using the same protocols as the day experiment, but each treatment ran for 16 hours between 3.30 pm and 7.30 am. It was necessary to use a ‘block’ design in this study because the fish community changed each day as new fish approached the regulator seeking a migration route. Thus, when treatments were performed, there could have been a different population of fish every day.’

Please note that the use of a randomized block design is commonplace in fish passage research and has been applied to many previously-published works (e.g. Baumgartner et al. 2019; Mallen-Cooper et al. 2008; Stuart et al. 2008).

Please also see our response to comment 16 by Reviewer 3.

Reviewer 3 Report

The authors present an interesting study on fish passage of small, tropical fish in the context of river – floodplain movement, through a relatively new type of fishway (cone fishway). The system, the species and the type of fishway is little research so the paper offers important new knowledge. The paper is well written. I have, however, have a few comments and questions.

My main concerns are:

During the experiments, fish were not prevented from entering the fishway from above (the exit of the fishway)? How do you know that fish present by the exit of the fishway have ascended the fishway and didn’t just drop down from upstream? Do you have any data on fish entering the fishway from upstream (through the exit of the fishway) to be able to say if it is unlikely that they did so, or that it is unlikely that a large proportion of fish present by the exit entered from above the fishway? This question is fundamental to the interpretation of the results of the study. I lack a discussion/introduction to the ecology of river – floodplain migration/movement. Why are these fish moving from the river to the floodplain and back again? Feeding, reproduction, escape predation by entering the floodplains? Escape drought, high temp, predation in the dry season by going down to into the main river? Going back after having been displaced by floods? What is known? If nothing else, with a few examples. I think it deserves a paragraph in the introduction. It is not the focus of the paper, but I would still like to see some discussion on the potential problems for fish finding the entrance and entering the fishway. Otherwise the relative successful ascent of many fish species might add to the myth that connectivity problems are easily fixed with fishways. Elsewhere, there are many fish passage solutions where the fish are able to pass the fishway, but the fish passage solution does not work because fish will not find or enter the fishway.

Here follows additional comments and questions:

L24-25: “Fisheries” or “fish populations”. I guess both, but “Fisheries” typically don’t go extinct?

L44: Write out RFSs. The abbreviation does not occur that often, and it would spare many readers from browsing back to see what it means.

L48: Complete life cycles?

L51: Write out the abbreviation.

L61-62: Such fishways should pose a similar problem also outside developing countries.

L67: Remove “bespoken”

L69: “Rock ramp design principles” – could you expand on what these principles are?

L99, L109, L112, L118: Write out the abbreviation, LMB -> Lower Mekong Basin.

L108: Remove “regarded as”. By any definition, it is one of the largest rivers.

L117-124: Study site: I’m unfamiliar with this kind of structure. What is the main purpose of the floodplain regulator? To save water for irrigation? Or store water for hydropower production downstream? Something else?

Figure 1 (but maybe amended in text and not in the figure): What is the expected route of the fish? Mekong River -> Flood runner -> Fishway -> Wetland/floodplain? Before the floodplain regulator was it Mekong River -> Floodplain? Is the Flood runner also a consequence of the impoundment?

L135: “100-250 mm wide slots.” Does this hinder larger fish from passing? Or are there few such large fish performing this river-floodplain/wetland movement/migration?

L180-197: One block consists of 1 day and 1 night trial? Treatment is day/night? One experiment is one trial? Maybe try using just trial and day/night instead of confusing the reader with “experiment” and “treatment”?.

L192-193: “Before commencing an experimental block for each experiment, the fishway was flushed of all fish by implementing a high flow continuously for a period of 30 minutes.” Do you have any idea about how efficient this is for removing the fish from the fishway?

L201-202: “Fishway location” sounds a lot like where the fishway itself is placed. Maybe “Location in the fishway (by entrance or exit)”?

L217: Remove “day and dusk/night”, it is only confusing. You start thinking about what is not analyzed while it is all biomass values that are analyzed.

L225: Here I’m confused what “experiment” is. It is day/night right? But then what is treatment? Goes back to my comment about experiment/treatment/trial above.

Methods: Could lower biomass by the exit to some extent be caused by fish not having reached the exit yet, but being in ascent. For the entrance, the fish is available to be caught immediately at entry but to be able to be caught by the exit, the fish must first ascend the fishway. Contrary to this, fish being able to ascend the fishway relatively quickly but being delayed at the exit (due to high water velocities, or just reluctance to transition into another environment) would artificially increase the biomass and fish numbers by the exit, right? Maybe these methodological questions are something that you could discuss in the discussion? To somehow follow individual fish could tease these things apart (but is of course not always feasible).

 L247: “Passage efficiency” should be defined in Methods. Especially since it is here used in a different – but related - manner to what is often the case in the fish passage literature (individual fish passing / individual fish attempting to pass).

L252: Remove “however”. This result is inline with what to expect and what is said in the previous line. The “however” makes you think that the result points in another direction. I also think that the results would read easier if the same location (maybe preferably the entrance) was always the base. So lower, longer, shorter, higher are always in relation to the numbers at the entrance.

L260-262: More occurrent by the exit than by the entrance. How is that explained? Delay before exiting the fishway? Or they entered from above?

L300: “Yet” indicates a contradiction from the previous statement, but shorter fish by the exit is actually in line with expected/previous? See comment above, for day-time data.

L339-343: These numbers are not very impressive (no fault of the study of course!). If the fish are reproducing or growing on the floodplains that would mean that the numbers means more fish than they actually say. Do we have any idea of the extent of the natural fish movement at the location that the fishway is supposed to partially restore? Maybe something to discuss?

L354: In this case, maybe you should add “4) Exit the fishway” to show what you’re not studying here. Especially in relation to the relative success of the fishway in relation to fish ascending it, it is important to discuss what elements of the fish passage process that haven’t been studied (see one of my initial comments).

L356: You don’t have the proportion of the fish that entered the fishway, only a sample of the number that did so.

L364: Can you elaborate a little on the Australian results?

L364-366: “We also noted a greater overall abundance of fish, and diversity of species and sizes than vertical slot and submerged orifice fishway designs tested at Pak Peung in the preceding year”. Do you think this is due to the location of the fishway (in relation to fish abundance and/or hydraulic conditions making the entry efficiency higher)? Or do you think this fish way somehow is intrinsically easier to enter? In that case why?

L376-384: Doesn’t these black fish have a Latin name? Are their just black fish, or is black fish their name?

L377-378: Floodplain and wetland, here it is the same habitat, right? Maybe make that clear somewhere in the intro. Or just stick to using one of the words?

L380-383: Why incidental? This brings me back to the question about why fish wants to go from the river to the floodplain.

L385-395: I’m not sure this is needed in the Discussion. It does not really concern the results of the study?

L438-445: I think the implication should focus on the study and not other (potential) advantages of the fishway. Or at least also discuss other aspects of passing fish that were not studied here. Like finding and exiting the fish way. The paper shouldn’t oversell the fishway.

Author Response

Reviewer 3

The authors present an interesting study on fish passage of small, tropical fish in the context of river – floodplain movement, through a relatively new type of fishway (cone fishway). The system, the species and the type of fishway is little research so the paper offers important new knowledge. The paper is well written. I have, however, have a few comments and questions.

My main concerns are:

1.During the experiments, fish were not prevented from entering the fishway from above (the exit of the fishway)? How do you know that fish present by the exit of the fishway have ascended the fishway and didn’t just drop down from upstream? Do you have any data on fish entering the fishway from upstream (through the exit of the fishway) to be able to say if it is unlikely that they did so, or that it is unlikely that a large proportion of fish present by the exit entered from above the fishway? This question is fundamental to the interpretation of the results of the study.

Authors’ response

Yes you are correct in that downstream migrating fish do use the fishway. We did not prevent downstream migrating fish on the assumption that these fish can move down the fishway unimpeded. When we were trapping, the traps prevent downstream migrating fish from entering the trap unless these fish turned around and then ascended back upstream and this seems unlikely. While a small number of fish may have made localised downstream movements into the fishway — then after a partial descent turned around and gone back upstream — and been caught in our trap, we are confident that this would be a very small number compared to those actively migrating upstream. While we cannot say that our samples are exclusively upstream migrating fish, our experience is that catches of downstream migrating fish — that then turned around and went upstream — are very much incidental. This is because we have performed extensive downstream passage surveys at the site, and most fish pass through the regulator itself (which has a higher discharge) and not the fishway. These results are presently being worked into a separate manuscript for publication later this year. This was a good comment thank you.

We have added the following text to the methods to explain why it is highly unlikely that any of the fish in the exit trap had come from upstream of the fishway:

‘These methods were tailored to catch upstream migrating fish (Baumgartner et al. 2019), and there were no observations of fish entering the fishway traps from other sources, such as downstream migrating fish from the wetland.’

2.I lack a discussion/introduction to the ecology of river – floodplain migration/movement. Why are these fish moving from the river to the floodplain and back again? Feeding, reproduction, escape predation by entering the floodplains? Escape drought, high temp, predation in the dry season by going down to into the main river? Going back after having been displaced by floods? What is known? If nothing else, with a few examples. I think it deserves a paragraph in the introduction.

Authors’ response

We have added more introductory content on the migration ecology of Mekong River fish species. Specifically, we have replaced:

The Mekong River has one of the most productive and diverse freshwater fisheries in the world [17] but is currently under immense pressure from widespread irrigation and hydropower development [2, 5]. The 4800-km long river supports over 60 million people throughout six Asian countries (China, Myanmar, Lao PDR, Thailand, Cambodia and Vietnam [18], and is home to an estimated 2000 species of fish [17]. The fishery in the Lower Mekong Basin (LMB — the portion of the basin located in Lao PDR, Thailand, Cambodia and Vietnam) provides between 48% (for Laos) and 79% (for Cambodia) of animal protein intake, respectively [18]). Annual yield from the capture fishery forms approximately 2% of the world’s total marine and freshwater catch [6]; consequently, the Mekong River fishery is crucial for supporting the food requirements and incomes of the neighbouring human populations in the LMB [6]. There is a critical need to develop and refine innovative fishway technologies to ameliorate the impacts of the proliferation of physical barriers in the LMB.’

with:

‘The Mekong River has one of the most productive and diverse freshwater fisheries in the world [17] but is currently under immense pressure from widespread irrigation and hydropower development [2, 5]. The 4800-km long river supports over 60 million people throughout six Asian countries (China, Myanmar, Lao PDR, Thailand, Cambodia and Vietnam [18], and is home to an estimated 2000 species of fish [17]. The fishery in the Lower Mekong Basin (LMB — the portion of the basin located in Lao PDR, Thailand, Cambodia and Vietnam) provides between 48% (for Laos) and 79% (for Cambodia) of animal protein intake, respectively [18]). Annual yield from the capture fishery forms approximately 2% of the world’s total marine and freshwater catch [6]; consequently, the Mekong River fishery is crucial for supporting the food requirements and incomes of the neighbouring human populations in the LMB [6].

Forty to seventy percent of fish catch in the Mekong consists of species that regularly undertake longitudinal and/or lateral migrations to access spawning, nursery, feeding or refuge habitat (Barlow et al. 2008). For instance, some of these species migrate from the main channel to floodplain wetlands and lakes during the wet season to access nursery habitat [6], while others live in floodplain wetlands and lakes during the wet season and migrate to tributaries or the main channel during the dry season [19]. Consequently, there is a critical need to develop and refine innovative fishway technologies to ameliorate the impacts of the proliferation of physical barriers in the LMB.’

3.It is not the focus of the paper, but I would still like to see some discussion on the potential problems for fish finding the entrance and entering the fishway. Otherwise the relative successful ascent of many fish species might add to the myth that connectivity problems are easily fixed with fishways. Elsewhere, there are many fish passage solutions where the fish are able to pass the fishway, but the fish passage solution does not work because fish will not find or enter the fishway.

Authors’ response

Given the newness of the cone fishway, there are still many aspects of the design that are yet to be empirically assessed. Its entrance attraction efficiency is one such aspect. While it was beyond the scope of our current study, we argue that this should be targeted as a high priority for investigation in future studies. We think that the reviewer has a good point and therefore, in the Discussion we have added:

‘While further work is needed to evaluate the efficiency of the fishway entrance in attracting migratory fish, high numbers of LMB fish, comprised of a diverse range of species and size classes were able to both locate and enter the bottom of the fishway, and fully ascend during both the day and dusk/night experiments (e.g. Parambassis siamensis, Sikukia gudgeri, and Xenentodon sp.).’

In addition, after ‘For example, it may be possible to improve the suitability of the cone fishway design for small-bodied LMB fish by applying more conservative design criteria such as a reduced slope, decreased discharge, or greater pool size [12], although such modifications increase fishway length and cost, and may adversely influence the efficacy for large-bodied species.’ we have added the following:

‘Future research should attempt to assess the impacts of these modifications on the fishway’s entrance attraction efficiency and the migratory community in general (biomass, species composition, size structure) — each within the context of the broader downstream community (Stuart and Marsden 2019).’

4.L24-25: “Fisheries” or “fish populations”. I guess both, but “Fisheries” typically don’t go extinct?

Authors’ response

We would prefer to use ‘fisheries’ in this context if we had to choose between the two terms. We are not trying to say that they may go extinct, but instead that they ‘are under threat’ and thus may be diminished over time if the threat continues to persist or becomes worse. That said, after construction of the regulator, the locals report that many species became locally extirpated upstream of the new barrier. Many of these species have returned now that the fishway has been completed (see Millar et al 2018).

5.L44: Write out RFSs. The abbreviation does not occur that often, and it would spare many readers from browsing back to see what it means.

Authors’ response

We have written out ‘river-floodplain systems’ and removed ‘RFSs’ throughout the manuscript, as requested. Thanks for the suggested improvement.

6.L48: Complete life cycles?

Authors’ response

We have replaced ‘subsequently hampered the completion of life cycle stages’ with ‘subsequently prevented fish from completing their life cycles’. Thanks for the suggestion.

7.L51: Write out the abbreviation.

Authors’ response

Addressed. Please see our response to comment 5 by Reviewer 3.

8.L61-62: Such fishways should pose a similar problem also outside developing countries.

Authors’ response

We have replaced: ‘In developing countries with lower construction standards’ with ‘In countries with low construction standards’.

9.L67: Remove “bespoken”

Authors’ response

This term was removed as requested.

10.L69: “Rock ramp design principles” – could you expand on what these principles are?

Authors’ response

Sentence modified to include more explanation in parentheses:

‘The cone fishway is based on rock ramp design principles (i.e. where there are multiple ascent pathways rather than a single slot), and……’

11.L99, L109, L112, L118: Write out the abbreviation, LMB -> Lower Mekong Basin.

Authors’ response

This acronym is internationally accepted and has been used in numerous publications (e.g. Grumbine and Xu 2011; Piman et al. 2013; Chea et al. 2016). We have sufficiently explained it upon its first mention in the text, so we believe that it should be kept in for the sake of efficiency.

12.L108: Remove “regarded as”. By any definition, it is one of the largest rivers.

Authors’ response

Now deleted as requested. Thanks for the suggestion.

13.L117-124: Study site: I’m unfamiliar with this kind of structure. What is the main purpose of the floodplain regulator? To save water for irrigation? Or store water for hydropower production downstream? Something else?

Authors’ response

A floodplain regulator refers is a physical structure used to regulate water movement through a floodplain water barrier such as a weir, usually either for flood protection or water security purposes (irrigation, hydropower generation etc.). At Pak Peung, the regulator is being used to secure water for irrigation during the dry season, and to prevent the rice crops from being inadvertently flooded during the wet season. We will clarify this by making the following changes to the text.

We have replaced ‘The regulator’s sluice gates prevent the village’s floodplain crops from being inadvertently flooded when wet season water levels rise’ with:

‘The regulator’s sluice gates prevent floodplain rice crops from being inadvertently flooded when wet season water levels rise, and allow for improved water security to support irrigation during the dry season.’

14.Figure 1 (but maybe amended in text and not in the figure): What is the expected route of the fish? Mekong River -> Flood runner -> Fishway -> Wetland/floodplain? Before the floodplain regulator was it Mekong River -> Floodplain? Is the Flood runner also a consequence of the impoundment?

Authors’ response

You are correct. The expected route of the fish is Mekong River -> flood runner -> fishway -> wetland.

The flood-runner is a natural watercourse and not a consequence of the impoundment. Prior to the installation of the sluice gate regulator, fish were able to move from the Mekong up the flood-runner into the wetland.

To explain this more clearly, we have replaced the last sentence in the second paragraph of the Methods with:

‘Prior to the installation of the regulator, fish could move upstream from the Mekong River to Pak Peung wetland via the flood-runner. However, the installation of the regulator has resulted in upstream fish passage being totally obstructed, and fish only being able to move downstream through the sluice gates, from Pak Peung wetland into the Mekong River, when the gates are open [6]. Locals report that many species disappeared from the wetland following regulator construction (Millar et al. 2018).’

Please also see our response to comment 4 by Reviewer 1.

15.L135: “100-250 mm wide slots.” Does this hinder larger fish from passing? Or are there few such large

Authors’ response

Good question. We have added a sentence to clarify:

‘This design arrangement was primary targeted toward passage of fish <0.6 m long, but larger fish (e.g. whiprays up to 1 m long) have ascended similar cone fishways in tropical Australia (Stuart and Marsden 2019).’

16.L180-197: One block consists of 1 day and 1 night trial? Treatment is day/night? One experiment is one trial? Maybe try using just trial and day/night instead of confusing the reader with “experiment” and “treatment”?.

Authors’ response

The day experiment and dusk/night experiment were treated totally independently because they were undertaken over different durations (this is why the results for each experiment have been completely separated).

For the day experiment, there was one treatment (location in the fishway: entrance vs. exit), and the sampling for each experimental block was conducted over two consecutive days (i.e. for each block, the entrance was sampled on one day and the exit was sampled on the other day in a randomised order).

Similarly, for the dusk/night experiment, there was one treatment (location in the fishway: entrance vs. exit), and the sampling for each experimental block was conducted over two consecutive nights (i.e. for each block, the entrance was sampled on one night and the exit was sampled on the other night in a randomised order).

We have made the following changes in the methods to make this clearer:

In the first paragraph within the Experimental design, we have replaced ‘two separate randomised block experiments’ with ‘two independent randomised block experiments’. In the second paragraph within the Experimental design, we have replaced ‘Both experiments’ with ‘Both the day and dusk/night experiments’. Please also see our response to comment 2 by Reviewer 2.

17.L192-193: “Before commencing an experimental block for each experiment, the fishway was flushed of all fish by implementing a high flow continuously for a period of 30 minutes.” Do you have any idea about how efficient this is for removing the fish from the fishway?

Authors’ response

Addressed. Please also see our response to comment 1 by Reviewer 2.

18.L201-202: “Fishway location” sounds a lot like where the fishway itself is placed. Maybe “Location in the fishway (by entrance or exit)”?

Authors’ response

We have clarified the text explaining the experimental design (please see our response to comment 2 by Reviewer 2 and comment 16 by Reviewer 3). Thanks for the improvement.

19.L217: Remove “day and dusk/night”, it is only confusing. You start thinking about what is not analysed while it is all biomass values that are analysed.

Authors’ response

We have replaced ‘The day and dusk/night fish biomass values’ with ‘The fish biomass values’. Thanks for the improvement.

20.L225: Here I’m confused what “experiment” is. It is day/night right? But then what is treatment? Goes back to my comment about experiment/treatment/trial above.

Authors’ response

We have clarified the experimental design more in the Methods to reduce the potential for any confusion. Please also see our response to comment 2 by Reviewer 2 and comment 16 by Reviewer 3.

We have also clarified this particular statement more by replacing ‘separately for each experiment’ with ‘separately for the day and dusk/night experiments’.

21.Methods: Could lower biomass by the exit to some extent be caused by fish not having reached the exit yet, but being in ascent. For the entrance, the fish is available to be caught immediately at entry but to be able to be caught by the exit, the fish must first ascend the fishway. Contrary to this, fish being able to ascend the fishway relatively quickly but being delayed at the exit (due to high water velocities, or just reluctance to transition into another environment) would artificially increase the biomass and fish numbers by the exit, right? Maybe these methodological questions are something that you could discuss in the discussion? To somehow follow individual fish could tease these things apart (but is of course not always feasible).

Authors’ response

We had touched on this potential issue in the Diurnal Patterns section of the Discussion, but we have now expanded the Discussion by suggesting ways for future studies to investigate this issue through assessments of individual fish movement patterns.

After the sentence ‘In long fishways, such as at Pak Peung, fish with strong diurnal movement preferences need to be able to complete their ascent during their preferred diurnal period, and further research is needed to clarify whether this is a limiting criterion.’, we have added:

‘These patterns could potentially be investigated in future studies by following the movements of individual fish throughout the fishway using a mark-recapture approach such as PIT tagging (Castro-Santos et al. 1996).’

 22.L247: “Passage efficiency” should be defined in Methods. Especially since it is here used in a different – but related - manner to what is often the case in the fish passage literature (individual fish passing / individual fish attempting to pass).

Authors’ response

We have now defined passage efficiency in the methods.

In the first paragraph of the Data Analysis section, prior to the sentence ‘The CPUE and BPUE datasets were initially square root (x+0.5) transformed to normalise their distributions and homogenise their variances, but the species richness and length (overall and common species) datasets were left untransformed because they were already homoscedastic and normally distributed.’, we have added the following sentence:

‘Passage efficiency was also assessed by considering the total abundance of fish at the fishway exit (averaged across all species) as a percentage of the total abundance of fish at the fishway entrance (averaged across all species).’

23.L252: Remove “however”. This result is inline with what to expect and what is said in the previous line. The “however” makes you think that the result points in another direction. I also think that the results would read easier if the same location (maybe preferably the entrance) was always the base. So lower, longer, shorter, higher are always in relation to the numbers at the entrance.

Authors’ response

‘However’ has been removed as requested. We have also switched the comparisons (where necessary) so that they are all in relation to the numbers at the entrance.

24.L260-262: More occurrent by the exit than by the entrance. How is that explained? Delay before exiting the fishway? Or they entered from above?

Authors’ response

It is reasonably common for more fish to appear at the exit than the entrance in prior fishway studies (i.e. Stuart and Mallen-Cooper 1999). This usually occurs because a species has a strong diurnal movement pattern and so fish aggregate in the resting pools until their preferred diurnal period when they continue their ascent and exit in significantly greater numbers than they enter at the bottom. It also occurs as a result of the schooling nature of some species.

In the Diurnal Patterns paragraph in the Discussion, we have added the following sentence prior to the last sentence:

‘Similarly to that observed for individual species in other paired fishway entrance/exit studies (e.g. Stuart and Mallen-Cooper 1999), several species (Hypsibarbus lagleri, Amblyrhynchichthys micracanthus and Barbonymus altus) occurred in a greater proportion of exit samples than entrance samples during our day experiment, possibly because these species aggregated in the resting pools before continuing their ascent, and subsequently exited in greater numbers than they entered at the bottom.’

25.L300: “Yet” indicates a contradiction from the previous statement, but shorter fish by the exit is actually in line with expected/previous? See comment above, for day-time data.

Authors’ response

 ‘Yet’ has been removed as requested. We have also again switched the comparisons (where necessary) so that they are all in relation to the numbers at the entrance.

26.L339-343: These numbers are not very impressive (no fault of the study of course!). If the fish are reproducing or growing on the floodplains that would mean that the numbers means more fish than they actually say. Do we have any idea of the extent of the natural fish movement at the location that the fishway is supposed to partially restore? Maybe something to discuss?

Authors’ response

Unfortunately there are no data quantifying the movement of fish between the Mekong River and Pak Peung wetland prior to the installation of the fishway at that site, so we cannot comment directly on that aspect. However, we have previously published work on the behaviour of fishers (Millar et al. 2018) which demonstrates that the area downstream of the regulator was very popular for fisherman suggesting that fish accumulations were occurring.

We agree that the biomass for Pak Peung does not seem very impressive when compared to the biomasses transferred by fishways in other countries and/or settings (e.g. longitudinal migrations on the Parana River in Argentina — Oldani and Baigun 2002). Nonetheless, as you’ve also pointed out, this biomass is likely to be greatly increased if any of the fish go on to reproduce and/or grow in the wetland. Further, the annual increases in biomasses need to be multiplied by the expected life of the fishway, which is 40–50 years (Crase et al. 2019). The cumulative effects of multi-year passage will yield significant benefits to the fishery. We had already touched on this aspect in the Discussion (see below), but we have now expanded our consideration.

Specifically in the Discussion paragraph starting with ‘Fish biomass transfer….’, we have replaced: ‘In addition, a number of small black fish species’ (i.e. species’ that typically only live in floodplain lakes and wetlands) ascending the fishway were observed to be ovigerous (i.e. carrying eggs), suggesting that these fish were migrating to spawn in the wetland, further augmenting productivity [28].’

with:

‘In addition, a number of small black fish species’ (i.e. species’ that typically only live in floodplain lakes and wetlands) ascending the fishway were observed to be ovigerous (i.e. carrying eggs), suggesting that these fish were migrating to spawn in the wetland, likely further augmenting productivity [29]. Indeed, the relatively moderate biomass of fish transferred to Pak Peung wetland may result in greater fish biomass within the wetland over the course of several years, via spawning, growth and recruitment.’

We have also added the following paragraph prior to the last paragraph in the Management Implications section:

‘The optimal solution to restoring fish passage at any site is to remove the physical barrier causing the issue. However, given that this is not possible at Pak Peung, the cone fishway provides an excellent alternative, and has the potential to support a marked increase in wetland fish productivity during the period of the fishway’s life time (40–50 years). This is a far better outcome than having no fishway, and thus no fish movement between the river and the wetland.’

27.L354: In this case, maybe you should add “4) Exit the fishway” to show what you’re not studying here. Especially in relation to the relative success of the fishway in relation to fish ascending it, it is important to discuss what elements of the fish passage process that haven’t been studied (see one of my initial comments).

Authors’ response

Thanks for the suggestion. We have added ‘fourthly, exit the fishway’ to the end of the sentence as requested.

Correspondingly, we changed ‘The current study involved undertaking day and dusk/night experiments to assess the latter two aspects…’ to ‘The current study involved undertaking day and dusk/night experiments to assess the second and third aspects…’.

28.L356: You don’t have the proportion of the fish that entered the fishway, only a sample of the number that did so.

Authors’ response

We did not word this very clearly — thanks for picking this up. It was meant to say ‘that is, the percentage of entrance-finding fish that could ascend the full length of the fishway (i.e. passage efficiency)’. We have now rephrased this accordingly.

Please also see our response to comment 3 by Reviewer 3.

29.L364: Can you elaborate a little on the Australian results?

Authors’ response

We have added in the following content to the end of the Discussion paragraph starting with ‘Despite the overall success of the cone design and passage of fish…..’

‘Indeed, Stuart and Marsden (2019) evaluated the effectiveness of cone fishways in the Norman, Fitzroy and Maribyrnong rivers, where they were effective in passing a broad suite of the local fish community, including fish as small as 9 mm long.’

30.L364-366: “We also noted a greater overall abundance of fish, and diversity of species and sizes than vertical slot and submerged orifice fishway designs tested at Pak Peung in the preceding year”. Do you think this is due to the location of the fishway (in relation to fish abundance and/or hydraulic conditions making the entry efficiency higher)? Or do you think this fish way somehow is intrinsically easier to enter? In that case why?

Authors’ response

We have clarified our text to show that we do not believe there is any inherent attraction advantage of the cone fishway just that this design was permanent rather than a temporary model, as shown below:

‘We also noted a greater overall abundance of fish, and diversity of species and sizes than in temporary experimental vertical slot and submerged orifice fishway designs tested at Pak Peung in the preceding year, although this was likely due to the permanent nature of outflow from the cone fishway than any other inherent attraction advantage [11].’

31.L376-384: Doesn’t these black fish have a Latin name? Are their just black fish, or is black fish their name?

Authors’ response

Within the context of the Mekong River Basin, black fish’ refers to various species of fish that typically only live on the floodplain (and not to a particular species).

We have made this clearer by changing ‘In addition, a number of small black fish (i.e. species that typically only live on the floodplain)’ to:

‘In addition, a number of small black fish species’ (i.e. species’ that typically only live in floodplain lakes and wetlands)’.

32.L377-378: Floodplain and wetland, here it is the same habitat, right? Maybe make that clear somewhere in the intro. Or just stick to using one of the words?

Authors’ response

We have now clarified this to include wetlands. Please see our response to comment 31 by Reviewer 3.

33.L380-383: Why incidental? This brings me back to the question about why fish wants to go from the river to the floodplain.

Authors’ response

Yes, you make a good point. We have modified our sentence to clarify that this aspect as not ‘incidental’ but reflects a broad range of reasons as to why fish utilise fishways and that black fish species returning to the wetland might actually be a rich area for further research.

We have replaced:

“While the reproductive status of black fish in the present study was probably incidental and these fish were simply trying to move back to the wetland after being displaced, the observation, nonetheless, further reinforces the value of effective fishways in allowing fish to access their preferred habitats to complete obligatory lifecycle processes

with:

‘While the reproductive status of the black fish in the present study provides for an area for further study, these fish were likely moving back into the wetland after being displaced. This observation further reinforces the value of effective fishways in facilitating access for fish to access their preferred habitats for a variety of obligatory lifecycle processes. If the fishway was not present, these fish would have been permanently displaced into unsuitable habitat.’

34.L385-395: I’m not sure this is needed in the Discussion. It does not really concern the results of the study?

Authors’ response

We have moderated this paragraph by reducing the focus on the fishway’s benefits. The paragraph now reads:

‘The success of the cone fishway at Pak Peung was likely related to its relatively conservative internal hydraulics. In particular, the cone fishway had a lower average pool turbulence (i.e. energy dissipation factor) of 50 W m-3 (Cd = 0.6) and water velocity (maximum 1.33 m.s-1) than prior fishways in the region [11, 16]. This was a key, and deliberate, design consideration.’

35.L438-445: I think the implication should focus on the study and not other (potential) advantages of the fishway. Or at least also discuss other aspects of passing fish that were not studied here. Like finding and exiting the fish way. The paper shouldn’t oversell the fishway.

Authors’ response

Yes, we agree and have added text to alert the reader to the need for more work on entrance and passage efficiencies, as well as the biomass aspect we raised earlier. We have also moderated the discussion in other areas (see our response to comment 34 by Reviewer 3). Thankyou for the chance to clarify this aspect.

We have replaced:

 ‘Further context-specific field assessments of new cone fishways will be crucial to optimize future applications, refine design features, and optimize attraction and exchange of biota between rivers and floodplains. Migratory river fishes comprise a very large group of species which are now at unprecedented risk from infrastructure development [38]; effective fishways are needed to help preserve ecosystem integrity of large tropical systems globally.’

with:

‘Further context-specific field assessments of new cone fishways will be crucial to optimise future applications, refine design features, and optimize entrance attraction, passage efficiency and exchange of biota between rivers and floodplains. There is also a need to ensure that cone fishways can adequately pass the high fish biomasses often found in tropical rivers. Fish passage is an adaptive science. Lessons are learned at new sites and applied to improve future projects. The information collected at Pak Peung will serve as examples of how to achieve better fish passage at other sites where irrigation structures are expected to impact fisheries productivity. Migratory river fishes comprise a very large group of species which are now at unprecedented risk from infrastructure development [38]; effective fishways are needed to help preserve ecosystem integrity of large tropical systems globally.’

Round 2

Reviewer 3 Report

I'm looking forward to seeing the paper published.